# A unified model library maps how neuromodulation reshapes the excitability landscape of neurons across the brain

Domenico Guarino[1,2]☯, Ilaria Carannante[1]☯*, Alain Destexhe[1]

**1** Department for Integrative and Computational Neuroscience (ICN), Paris-Saclay Institute of Neuroscience (NeuroPSI), Gif-sur-Yvette, France, **2** HABS, Institut des Neurosciences de Paris Saclay, Centre CEA Paris Saclay, France

☯ These authors contributed equally to this work.
* ilaria.carannante@gmail.com

## Abstract

The activity of neurons in the central nervous system changes drastically across different states like asleep or awake, attentive or drowsy, behaving or resting. Neuromodulation supports these changes by altering neuronal excitability. Much work has been devoted to understanding the effects of neuromodulators on various neurons. However, we still lack a cohesive picture of how neuromodulation affects neuronal dynamics. Here, we provide an analytical framework to characterise neuromodulation.

First, we analyse electrophysiological data from published papers and extract features for seven types of neurons from five areas of the human and rodent central nervous system, under the effect of five neuromodulators. Second, we describe this data using the widespread, simple, yet biologically accurate, Adaptive Exponential Integrate-and-Fire model (AdEx). Third, we use a dimensionality reduction technique to study the parameter space of the control and neuromodulated conditions. Fourth, we examine the dynamics of each neuromodulator through bifurcation and phase-plane analyses.

Our analyses revealed that: (i) different neuromodulators remap the parameters space of neurons into non-overlapping clusters, (ii) changes in electrophysiological properties induced by neuromodulators can be explained by transitions between dynamical regimes in the excitability landscape of neurons, and (iii) there are only two distinct neuromodulatory effects: "switching" – changing the spiking behaviour – and "scaling" – strengthening or weakening an existing behaviour. Leveraging this framework, we estimated the effect of neuromodulation in one species using data from another species. This study provides an integrative perspective on neuromodulation, offering experimenters and theoreticians a compact description of its impact on neuronal activity. The open-source Python-based workflow is modular and readily applicable to other datasets.

**Data availability statement:** All the data and implemented codes are available here: https://github.com/Computational-NeuroPSI/Neuromodulation Details of the software and code availability are included in the manuscript.

**Funding:** Research supported by CNRS, Agence Nationale de la Recherche (BrainAct and FLAG-ERA and JTC ImpactCom projects), and the European Union (Human Brain Project H2020-945539 and Virtual Brain Twin project 101137289).

**Competing interests:** We have no conflicts of interest to disclose. We confirm that this work is original and has not been published elsewhere, nor is it currently under consideration for publication elsewhere.

## Author summary

We provide a framework to study neuronal dynamics under neuromodulation. We characterised and modeled the electrophysiological features of seven types of neurons across five brain regions in humans and rodent under the effects of five neuromodulators. Each neuromodulator reshapes the neuronal parameters space into distinct clusters. Bifurcation and phase plane analysis reveal that neuromodulation operates through two modes: "switching" or "scaling". We show how effects observed in one species can inform estimations in another. This work provides a concise framework for understanding neuromodulation, useful for experimenters and theoreticians. In addition, we developed a modular, Python-based workflow that can be readily applied to other datasets.

## Introduction

Neuromodulation continuously influences our nervous system, dynamically shaping neuronal activity to regulate various cognitive, emotional, and behavioural functions [1]. Neuromodulators exert a variety of effects on brain cells, including changing neuronal excitability, shaping short and long-term synaptic plasticity [2] and altering astrocytic calcium signalling [3]. Here, we study how neuromodulation influences the dynamical properties of neurons. Neuromodulators act through various endogenous mechanisms, such as the modulation of ion channel conductances, the activation or inhibition of second-messenger pathways, and alterations in membrane properties [2,4], creating a cascade of interpretability problems that motivated this study.

There are hundreds of specific receptors for molecules capable of altering membrane excitability. These are usually divided into: amino acids (such as Glutamate and GABA), peptides (like Oxytocin and other endorphins), monoamines (including Dopamine, Adrenalin, Serotonin, Histamine and Norepinephrine), purines (Adenosine triphosphate, Adenosine), gasotransmitters (Carbon monoxide, Nitric oxide), and Acetylcholine [6]. Another common manner of dividing these molecules is into neurotransmitters (such as Glutamate and GABA) and neuromodulators (such as Acetylcholine and Dopamine). Neuromodulators differ from neurotransmitters in that (i) they are typically expressed by specific groups of neurons, (ii) project diffusely throughout the nervous system, (iii) usually have longer-lasting effects, and (iv) they modulate postsynaptic neurons by altering their responses to neurotransmitters [5,6].

There have been several modeling works detailing the interaction between specific neuromodulators, their target channels, and their effects on multicompartmental membrane models and microcircuits [7,8]. In addition, several frameworks exist for optimising the parameter space of detailed Hodgkin-Huxley models [9–11]. However, the main problem is that we are still lacking a cohesive general picture of how different families of neuromodulators affect neuronal dynamics. Considering detailed ion channel-based models would lead to the impossibility of comparing different

neurons expressing different types and distributions of channels. Therefore, to consolidate our understanding of neuromodulation dynamics, and foster the desired cohesion at a manageable scale, we adopted, here, the simple yet dynamically rich and conductance-based Adaptive Exponential Integrate-and-Fire (AdEx) model [12,13].

First, we aggregated different sources in the literature concerning seven types of neurons from five areas of the human and rodent central nervous system under the effect of five types of neuromodulators. Second, we extracted key features from electrophysiological recordings illustrated in published papers, and we conducted a parameter space exploration to fit the AdEx model parameters. Third, we analysed how neuromodulation influences the parameters of the control models and applied a dimensionality reduction technique, specifically principal component analysis (PCA), to visualise the parameter space and evaluate the formation of independent clusters. Finally, we studied the bifurcation and phase plane of this two-dimensional system, identifying two main types of neuromodulatory effects: "switching" – changing the spiking behaviour – and "scaling" – strengthening or weakening the same spiking behaviour. In addition, if two species exhibit comparable baseline excitability and data on a neuromodulator's effect is available for only one, our method enables cross-species estimation.

Electrophysiological recordings often include only a few traces per neuron type and this increases the risk of overfitting. To address this issue and overcome this limitation, we propose using not a single best-fit model, but rather a set of models for each condition. This has a double advantage: when new data becomes available, additional constraints can be introduced to filter out models that no longer satisfy the fitting (without starting the entire process); and most importantly, the distribution of parameters provides a robust estimate of their values, with the average serving as a reliable indicator.

This work aims to help bridge the gap between the electrophysiological and computational communities by considering several neuromodulators and neuron types, in a simple computational neuron model, and providing a framework for representing and interpreting the effects of neuromodulation on neuron excitability in a concise, reproducible and accessible manner.

## Materials and methods

### Data collection and curation

We systematically screened the available literature using an articulated PubMed search query.

```
( <neuromodulator>[MeSH Terms] OR <neuromodulator>[Title/Abstract] ) AND
( electrophysiology[MeSH Terms] OR "patch clamp"[MeSH Terms] OR
  "voltage clamp"[MeSH Terms] OR "current clamp"[MeSH Terms] OR
  "intracellular recording"[MeSH Terms] ) AND
( neuron[MeSH Terms] OR neuron*[Title/Abstract] OR
  "brain slice"[MeSH Terms] )
```

As of August 2025, this search query gives several articles: 1115 for Acetylcholine, 96 for Adrenaline, 579 for Norepinephrine, 1174 for Dopamine, 970 for Serotonin, and 138 for Histamine. We screened these results with the requirement that the original article must contain at least one voltage trace of control and <neuromodulator> conditions for the same recorded neuron type.

Using this approach we were able to select 30 voltage traces from five brain regions, representing seven neuronal types, recorded in control conditions and under the influence of five different neuromodulators. An overview of the recordings included in our dataset is provided in Table 1.

The selected publications did not provide raw electrophysiological data files, however the traces were available as figures. In order to retrieve the data, we extracted the voltage traces from images using the tool WebPlotDigitizer [47]. While this approach is not ideal, it allowed us to reconstruct the traces with sufficient resolution for our analysis.

**Table 1.** **Dataset of electrophysiological recordings across brain regions, species, and neuromodulators.**

| Region | Species | Condition | Temp.(°C) | Ref. | Figure |
|---|---|---|---|---|---|
| Cortex | Human | Control | 35 ± 1 | [15] | Fig 1A |
| Cortex | Human | Control | 35 ± 1 | [15] | Fig 1B |
| Cortex | Human | Control | 35 ± 1 | [15] | Fig 1D |
| Cortex | Human | Histamine | 35 ± 1 | [15] | Fig 1A |
| Cortex | Human | Methacoline | 35 ± 1 | [15] | Fig 1B |
| Cortex | Human | Norepinephrine | 35 ± 1 | [15] | Fig 1C |
| Cortex | Human | Serotonin | 35 ± 1 | [15] | Fig 1D |
| Cortex | Rat | Control | 31–32 | [44] | Fig 3a1 |
| Cortex | Rat | Dopamine | 31–32 | [44] | Fig 3a2 |
| Striatum | Mouse | D1_control | 34–35 | [26] | Fig 4C |
| Striatum | Mouse | D2_control | 34–35 | [26] | Fig 4C |
| Striatum | Mouse | D1_dopamine | 34–35 | [26] | Fig 4C |
| Striatum | Mouse | D2_dopamine | 34–35 | [26] | Fig 4C |
| Striatum | Rat | Control | 34–36 | [27] | Fig 8A |
| Striatum | Rat | Acetylcholine* | 34–36 | [27] | Fig 8B |
| Hippocampus | Mouse | Control | 25 | [29] | Fig 5A (60pA) |
| Hippocampus | Mouse | Control | 25 | [29] | Fig 5A (100pA) |
| Hippocampus | Mouse | Control | 25 | [29] | Fig 5A (140pA) |
| Hippocampus | Mouse | Acetylcholine** | 25 | [29] | Fig 5A (140pA) |
| Hippocampus | Rat | Histamine | 30 | [30] | Fig 3 |
| Hippocampus | Rat | Norepinephrine*** | 30 | [31] | Fig 2A |
| Thalamus | Rat | TPS Control | 32 ± 1 | [33] | Fig 5Aa |
| Thalamus | Rat | TPS Orexin-B | 32 ± 1 | [33] | Fig 5Ac |
| Thalamus | Rat | TPS Norepinephrine | 35 ± 1 | [38] | Fig 2B |
| Thalamus | Rat | TPS Acetylcholine | 35 ± 1 | [38] | Fig 4A |
| Thalamus | Rat | TRN Control | 36 ± 1 | [41] | Fig 3A |
| Thalamus | Rat | TRN Acetylcholine | 36 ± 1 | [41] | Fig 3A |
| Cerebellum | Rat | Control | 31–33 | [42] | Fig 1F |
| Cerebellum | Rat | Norepinephrine | 31–33 | [42] | Fig 1F |
| Cerebellum | Rat | Acetylcholine* | 21–23 | [43] | Fig 6C |

\* Muscarine, a selective agonist of Acetylcholine was bath applied.
\*\* Oxo-M, a muscarinic agonist was bath applied.
\*\*\* Isoproterenol, a $\beta$-adrenergic agonist was bath applied.

All the extracted traces were saved in `.csv` format and are now available in the GitHub folder https://github.com/Computational-NeuroPSI/Neuromodulation/tree/main/data/data_traces.

During the data curation step, four features concerning the membrane voltage are measured: the initial value of the voltage, holding voltage; the value of the voltage reached during the spike, voltage peak ($V_{peak}$); the voltage corresponding to the initiation of the spike, voltage threshold ($V_{th}$); and after-spike reset potential ($V_{reset}$), as reported in S1 Table. The holding voltage measured in the absence of injected current is used to derive $E_L$. A metadata file including these values and current protocols was also created and made available in the same folder.

### Feature extraction

We developed Python scripts to load the `.csv` files and extract key electrophysiological features from each voltage trace. These features, together with the associated current injection protocols, were saved in `.json` files which we then used for our modeling and analysis. As anticipated, we focused on features which allow us to describe the overall behaviour of the traces and are compatible with the level of detail we can retain. These key features are time to first, second, third and last spike, inverse of the first and last interspike interval, and firing frequency.

From the metadata file, we read in the information about the current injection protocol. This includes the amplitude of the injected current (`curr`) as well as the temporal structure of the protocol: delay before stimulus onset (`stim_delay`),

the time at which the stimulus ends (`stim_end`) and the total recording duration, all expressed in ms. From these we can easily compute the `stim_duration` (`stim_end` - `stim_delay`) and proceed with the extraction of the features.

The core mechanism underlying the extraction is the *detection of the action potentials*. We do that by identifying when the membrane potential crossed a fixed voltage threshold $V_{th}$. This value depends on the neuron type and it is set by the user.

Specifically we initialize a `peak_index` list, iterate through the `trace_voltage` elements and if the following condition is verified, the corresponding index *i* is appended to `peak_index`:

$$\texttt{trace\_voltage}[i] > V_{th} \quad \text{and} \quad \texttt{trace\_voltage}[i-1] < V_{th}$$

In other words, a spike is detected when the voltage trace crosses the voltage threshold "from below".

From the `peak_index` list, we compute all the other features. The firing rate (`freq`) is computed as the number of spikes divided by the stimulus duration. The time to first spike (`time_to_first_spike`) is the delay between stimulation onset and the first detected peak. If at least two spikes are detected, we compute the time to second spike (`time_to_second_spike`) and the inverse of the first interspike interval (`inv_first_ISI`), which corresponds to the inverse of the difference between the first and second spike time. If at least three spikes are detected we compute the time to third spike (`time_to_third_spike`) and the inverse of the last interspike interval (`inv_last_ISI`). If more than three spikes are detected we also compute the time to last spike (`time_to_last_spike`). Finally, in cases where the traces exhibited strong spike-frequency adaptation (control dentate granule neurons, Fig 4 A; control thalamic relay neurons, Fig 5A), we have also included the value of the membrane potential at the end of the stimulus (`volt_stimend`) as an additional feature.

In the presence of strong spike-frequency adaptation using the total stimulus duration may underestimate the firing frequency. A finer measure could be to use instead the time to last spike. However, for the purpose of model fitting, the consistency of its computation across data and corresponding models is the key aspect. For this reason the total stimulus duration is used in all cases. We selected the inverse of the interspike interval rather than the interval itself to express these values in units consistent with frequency, making them easier to visualise and interpret in the context of firing behaviour.

A Jupyter Notebook file exemplifying this process can be found here Neuromodulation/extracting_features/Features_extraction.ipynb. We do not consider features describing the spike shape because we do not have access to the raw data and also the AdEx model is not designed to reproduce detailed spike waveforms. However, we further refine the models that best match the extracted features by visually inspecting the spike shapes they generate and we prefer those that qualitatively resemble the recorded traces the best. This step is not mandatory, but it allows us to ensure that even if not explicitly fitted, the spike shapes are reasonably reproduced within the limits of the data itself and the model.

### AdEx model

We use the Adaptive Exponential Integrate-and-Fire model (AdEx), introduced by [12], theoretically characterised by [13], and experimentally characterised for the cortex by [14]. This model is powerful enough to capture the rich and diverse dynamic of neurons under different neuromodulators. Despite the small number of parameters, it retains a clear biological meaning, and its computational efficiency makes it suitable for large-scale network simulations.

The AdEx describes the evolution of the membrane potential $V(t)$ of a single-compartment neuron. It consists of a system of two differential equations:

$$\begin{cases} C\dfrac{dV}{dt} = -g_L(V - E_L) + g_L \Delta_T \exp\left(\dfrac{V - V_{th}}{\Delta_T}\right) + I - w \\ \tau_w \dfrac{dw}{dt} = a(V - E_L) - w \end{cases} \tag{1}$$

When the potential goes beyond $V_{th}$, the exponential term generates an upswing of the membrane potential that mimics the exponential activation of $Na^+$ channels in a Hodgkin–Huxley-type neuron model, leading to the initiation of an action potential. At a spike detection (peak voltage $V_{peak}$, e.g. 0 mV), a series of reset conditions is triggered:

$$\begin{cases} V \to V_{reset} \\ w \to w + b \end{cases} \tag{2}$$

The increase of the adaptation variable $w$ by a value $b$ leads to accumulation during a spike train.

There are nine parameters required to define the evolution of the membrane potential ($V$) and the adaptation current ($w$), divided into *scaling parameters*: $C$ membrane capacitance, $g_L$ leak conductance, $E_L$ leak equilibrium potential, $\Delta_T$ exponential slope factor, $V_T$ effective threshold potential, and *bifurcation parameters*: $a$ subthreshold adaptation conductance, $b$ spike-triggered adaptation, $\tau_w$ adaptation time constant, and $V_{reset}$ after-spike reset potential. Modifying the bifurcation parameters yields qualitative changes in the behaviour of the system.

## Model simulation

A core mechanism of the algorithm is the simulation of the AdEx models, which consists in numerically solving the two coupled differential equations that describe the membrane potential and adaptation variable at each time step (Eq 1).

To do so, we implemented in Python the forward Euler numerical method using a default time step of 0.01 ms. To handle the spike discontinuity, if the membrane potential $V$ reaches or exceeds the peak voltage $V_{peak}$, $V$ is reset to $V_{reset}$ and $w$ is increased by $b$; then, for a refractory period (usually within 5 ms), both quantities are kept constant. While the forward Euler method is simple and less accurate than, for example, higher-order Runge-Kutta methods, its cheap computational cost and the smoothness of $V$ and $w$ before the peaks make it the preferred choice.

To provide a transparent and reproducible computational environment, we conducted all simulations using Python 3.9 and Jupyter Notebooks [55]. The modeling environment also utilized specialized libraries for numerical simulations (Numpy, version 1.21.6 [56]), data analysis (Scipy, version 1.8.1 [57]), and visualization(Matplotlib, version 3.5.2 [58].

To validate our implementation, we compared its solutions to those obtained using NEST and Brian 2, and confirmed that the results were qualitatively and quantitatively consistent. The models used for this comparison were initially taken from [14] (specifically, their Fig 4), which provided a useful starting point for testing the well-established simulators and our own implementation (S1 Fig). As the project progressed, we also extended the comparison to include the models generated using our pipeline. Both versions of the comparison are available as separate Notebooks here: https://github.com/Computational-NeuroPSI/Neuromodulation/tree/main/comparing_methods. It is important to note that each simulator uses its own conventions for units and proper unit conversion is therefore crucial to ensure consistency across implementations.

## Parameter exploration

The algorithm we developed takes as input the extracted features (reading them from the corresponding `.json` files) and the user-defined values/ranges for the AdEx model parameters. The spike peak voltage ($V_{peak}$) and the initial membrane potential (holding voltage) are fixed by the voltage traces. The ranges for $V_{th}$ and $V_{reset}$ are chosen based on the corresponding measured values. All remaining ranges are either kept at their default values or set based on prior knowledge and/or manual pre-tuning. Additionally, the user can specify the number of models to generate. We explore the parameter space by sampling each parameter within its specified bounds.

The next step consists in running the newly obtained models and extracting the features. These features are the input of the error function which quantifies the difference between the data-derived and model-derived features.

To efficiently sample the parameter space we use Sobol's sequences via the python library `scipy.stats.qmc.Sobol` [54]. We preferred it to the standard pseudorandom sampling because Sobol sequences are quasi-random low discrepancy which ensures a more uniform and space-filling coverage of the parameter space. Since Sobol sequences are built recursively using binary representations, as a number of models we select a number which can be expressed as a power of 2. To maintain this structure, when selecting the number of best models we chose 16 (i.e. $2^4$). So, with the goal of reducing clustering and gaps in the high-dimensional space we want to investigate we selected this method, however the user can easily change how the space is sampled and still be able to continue with the analysis.

## Model ranking

To evaluate how well each AdEx model reproduces the extracted features, we implemented an error function in Python (as part of the developed algorithm). This function, as anticipated, takes as input the values of each feature extracted from the data and the corresponding values extracted from the simulated models, and computes the relative error. It is defined as:

$$\texttt{error}[i,j] = \frac{\left|\texttt{data\_feature}[j] - \texttt{model\_feature}[i,j]\right|}{\left|\texttt{data\_feature}[j]\right|}$$

Where $i$ indexes the models and $j$ the features.

The total error for a given model is obtained by simply summing the relative errors across all features and all current protocols (if more than one is available). In cases where the model fails to "produce a feature" (for example, if it does not spike, many features can not be computed) we assign a penalty by adding +3 to the relative error for that specific feature. The value is arbitrary, it can of course be changed and the function can be personalized. Moreover, for example, in the cases where the traces to fit exhibited strong spike-frequency adaptation, the error in relation to the value of the membrane potential at the end of the stimulus was weighted more than those corresponding to other features. Users can adjust the relative weights of each feature in the error function to emphasize aspects of the data that are most relevant to their specific data traces. In the default configuration, all features are weighted equally.

Once the total error has been computed for each model, we rank all models in ascending order of error (by using `numpy.argsort`). We then store the top $n$ models, parameter set and error values, in a `.json` to allow the user to further analyze them.

An example demonstrating both the parameter exploration and model ranking procedure can be found here: https://github.com/Computational-NeuroPSI/Neuromodulation/tree/main/exploring_parameters_space.

In many studies, it is common practice to use a single AdEx model per neuron type and condition. However, in this work, we take a different approach by selecting a set of models (S2 Fig). During the parameter exploration and consequent fitting process, we often have access to only a single voltage trace, which may not be sufficient to constrain all the models in a way that ensures accurate reproduction of the *f-I* relation (when that curve is not available). By selecting the top $n$ models from our fitting procedure, we aim to capture a broader range of plausible dynamics, and reduce the risk of using a single model that performs well on a single trace but might fail to generalise the neuron behaviour.

## Post-fitting analysis

We performed four main post-fitting analyses for each neuron type.

**Comparing control and modulated AdEx parameters.** We compared the AdEx parameters obtained for control and modulated conditions by examining the distribution of each parameter (see S1 Appendix). Specifically, for each condition, we plotted the parameter values corresponding to the best 16 models and computed their mean and standard deviation. This allowed us to identify systematic shifts in the parameters space when neuromodulation was applied and also identify which parameters had a narrower range of variability. In this way we were able to spot the parameters which are more critical for capturing specific dynamics.

**Principal component analysis.** To visualize the structure of the parameter space and assess whether control and modulated models formed distinct groups, we performed a Principal Component Analysis on the best 16 models. The PCA revealed that models corresponding to different conditions formed clearly separate clusters in low-dimensional space and models fitted independently on traces corresponding to the same neuron and condition formed overlapped clusters. To quantify this separation, we computed the Silhouette score, which supported the reliability of the clustering. Both PCA and Silhouette score were computed using the scikit-learn library [61].

**Excitability landscape.**

**Excitability classes.** For every parameter set, we extracted:

$$R_a = \frac{a}{g_L}, \qquad R_\tau = \frac{\tau_m}{\tau_w}, \qquad \text{with } \tau_m = \frac{C_m}{g_L}.$$

Following the analytical classification of [13], the relative magnitudes of $R_a$ and $R_\tau$ uniquely determine the local bifurcation that terminates the resting state. In brief, a strong-and-fast adaptation brake ($R_a > R_\tau$) produces a sub-critical Hopf bifurcation (Class II onset), whereas a weak-or-slow brake ($R_a < R_\tau$) yields a saddle–node on invariant circle (Class I onset). Exact equality marks a codimension-2 Bogdanov–Takens (BT) point.

| Condition | First bifurcation | Functional signature |
|---|---|---|
| $R_a < R_\tau$ | **SNIC** (saddle–node on circle) | continuous *f-I* onset; arbitrarily low rate |
| $R_a > R_\tau$ | **Sub-critical Hopf** | rate jumps to finite value; short latency |
| $R_a = R_\tau$ | **Bogdanov–Takens** | boundary between the two regimes |

**Excitability landscape construction.** For each cell and condition, we plotted the point $(R_a, R_\tau)$ in the $R_a$–$R_\tau$ plane. Individual fits appear as small markers, and the centroid of control and neuromodulated conditions were joined by an arrow (control → modulated). The diagonal $R_a = R_\tau$ (dashed line in all figures) is the universal Class I/Class II boundary; no data-dependent scaling is applied. Large shifts that cross the boundary predict a qualitative regime change-continuous *f-I* curves with arbitrarily low rates, and longer first-spike latencies near rheobase - whereas smaller shifts keep models near the boundary and thus yield mixed signatures (e.g., shallow discontinuities or modest latency changes). The panel therefore provides a compact readout of how strongly each neuromodulator remaps the cell's excitability class and associated bifurcation mechanism. Throughout this work, we presented the excitability landscape, with model example phase planes (see below).

**"Switching" and "scaling".** The excitability landscape allows us to introduce a simplified nomenclature for movement around the excitability landscape. Neuromodulators can either **switch** a neuron's spiking behaviour, making it cross the $R_a = R_\tau$ diagonal from Class I (SNIC) to Class II (subcritical Hopf) and vice versa, or **scale** the onset bifurcation by modulating its expression. The Bogdanov-Takens (BT) boundary is given by:

$$\frac{a}{g_L} = \frac{\tau_m}{\tau_w} \quad \text{with} \quad \tau_m = \frac{C_m}{g_L} \implies a\tau_w = C_m.$$

Thus, excitability class depends on the product $a\tau_w$ relative to $C_m$ (and not on $g_L$): Class I (type-I/SNIC): $a\tau_w < C_m$ vs Class II (type-II/sub-Hopf): $a\tau_w > C_m$. Lowering $a\tau_w$ drives Class II → Class I transitions. Changes in $g_L$ alone move points roughly *along* the BT line (both coordinates scale together), leaving $a\tau_w$ unchanged. A Class I → Class II switch requires increasing $a$ and/or prolonging $\tau_w$ so that $a\tau_w > C_m$.

**Phase plane.**   Finally, since the AdEx model is described by two coupled differential equations, its dynamics can be easily visualised in a phase plane plot. To this end, we use the *V*-nullcline (Eq 3), which represents the set of points where the rate of change of *V* is zero (*dV/dt* = 0), and the *w*-nullcline (Eq 4), which consists of points where *dw/dt* = 0:

$$w = -g_L(V - E_L) + g_L \Delta_T \exp\left(\frac{V - V_{th}}{\Delta_T}\right) + I \tag{3}$$

$$w = a(V - E_L) \tag{4}$$

The intersection of the two nullclines defines fixed points – a specific combination of membrane potential (*V*) and adaptation current (*w*) at which both derivatives (*dV/dt* and *dw/dt*) are zero. We investigated the stability of those by analyzing the eigenvalues of the Jacobian matrix obtained from linearizing the system around the fixed point. If all eigenvalues have negative real parts, the fixed point is stable (attracting). If any eigenvalue has a positive real part, the fixed point is unstable (repelling). If there are both positive and negative eigenvalues, it is a saddle point, exhibiting stable and unstable directions (depending on the sign of the eigenvalue). This modelling approach allows the following intuitions. If the left branch of the V-nullcline during neuromodulation is less steep than the control condition, this means that the neuron requires less current to destabilise the resting state (*E_L*) and move toward spiking. In other words, the resting state becomes more sensitive to small perturbations. The slope of the *w*-nullcline encodes the parameter a (adaptation rate). When the adaptation is weaker and hence the slope lower, once the neuron starts firing there is less negative feedback to counteract depolarization. Hence, lower slope means higher excitability. Moreover, when looking at more depolarized membrane potentials, if the intersections between nullclines shift to the left, and therefore the distance to the bifurcation point (where the neuron transitions to spiking) is reduced, the neuron becomes closer to threshold and therefore more excitable. These effects shift the neuron toward easier spike initiation and more persistent firing.

### Estimating the effect of a neuromodulator

The AdEx model allows us to estimate the effect of neuromodulators across species, given that the baseline excitability properties of the control neurons are comparable. For example, we used rodent data to estimate the effect of dopamine on human cortical neurons, taking advantage of the fact that the slopes of the *f-I* curves of the corresponding control models are comparable (Fig 8F). To do so, two strategies were implemented. For the first strategy, *absolute value strategy*, we computed a *modulation ratio* for each parameter *i* as:

$$\texttt{ratio\_i} = \left| \frac{\texttt{avg\_rodent\_mod\_i}}{\texttt{avg\_rodent\_control\_i}} \right|$$

where `avg_rodent_mod_i` and `avg_rodent_control_i` represent the mean value of the *i*-th parameter under modulation and in control conditions, respectively. Then we apply these relative changes observed in rodent to the average human control values:

$$\texttt{estimated\_human\_mod\_i} = \texttt{avg\_human\_control\_i} \cdot \texttt{ratio\_i}$$

for each parameter *i* of the AdEx model.

However, if the corresponding control values of the two species do not have the same sign, the signed change is not preserved. This is for example the case of the parameter *a* (Fig 8C and S4A Fig).

In order to take that into account, a second strategy was implemented. In this case, the relative change for the *j*-th parameter which has different signs for the two species (in this case it is the parameter *a*) is computed as:

$$\texttt{ratio\_j} = 1 - \left| \frac{\texttt{avg\_rodent\_mod\_j}}{\texttt{avg\_rodent\_control\_j}} \right|$$

Then, analogously to the previous case, the *j*-th estimated value is computed as:

$$\texttt{estimated\_human\_mod\_j} = \texttt{avg\_human\_control\_j} \cdot \texttt{ratio\_j}$$

This approach reflects the same direction of change even if the baseline values have opposite signs (S4C Fig) and for this reason we refer to this as *signed value strategy*. Both strategies originate comparable results in terms of estimated dynamics, while the increase in firing frequency is more pronounced in the second case (S4B, D-F Fig). We have also tested alternative ways of scaling the parameter *a*. Those approaches resulted in values of *a* that fell between those obtained with the two strategies described above (see S1 Appendix and Jupyter Notebook). To further asses how variations in *a* effect the results, we performed a sensitivity analysis (S4E and F Fig), which confirmed that the qualitative results are preserved across this range of *a* values. A Notebook exemplifying these approaches can be found in the GitHub folder.

## Software and code availability

To ensure transparency and facilitate replication, we made our code (fully written in Python) and software publicly available on GitHub: https://github.com/Computational-NeuroPSI/Neuromodulation.

The repository contains a complete pipeline for analysing neuromodulatory effects on neuronal excitability using the AdEx model. It includes:

- Curated electrophysiological data derived from figures in published papers,
- Feature extraction tools to process the voltage traces
- AdEx simulation code using our implementation based on the forward Euler numerical method, NEST and Brian 2,
- Parameter space exploration and filtering of best models,
- Post-fitting analyses including parameter distribution comparisons, PCA, phase-plane, and excitability landscape plots,
- Estimation of neuromodulatory effects in species lacking direct data,
- Example scripts and Notebooks to reproduce results and visualise model behaviour.

Simulations were performed on a Linux-based desktop computer equipped with an AMD Ryzen 9 3900X 12-core (24-thread) processor and 64 GB of RAM.

## Results

Neuromodulators exert diverse effects on neurons. This variability hinders cross-study comparisons, and available data are often limited to a few electrophysiological traces. To enable comparison, we identified a minimal set of features from published traces in control and neuromodulated conditions: time to first, second, third, and last spike (red bars 1–4 in Fig 1A,B), inverse of the first and last interspike intervals (red bars 5–6), firing frequency, and, for strongly adapting neurons, voltage at stimulus end. These features are accessible, capture a broad range of dynamics, and are compatible with data extracted from diverse sources.

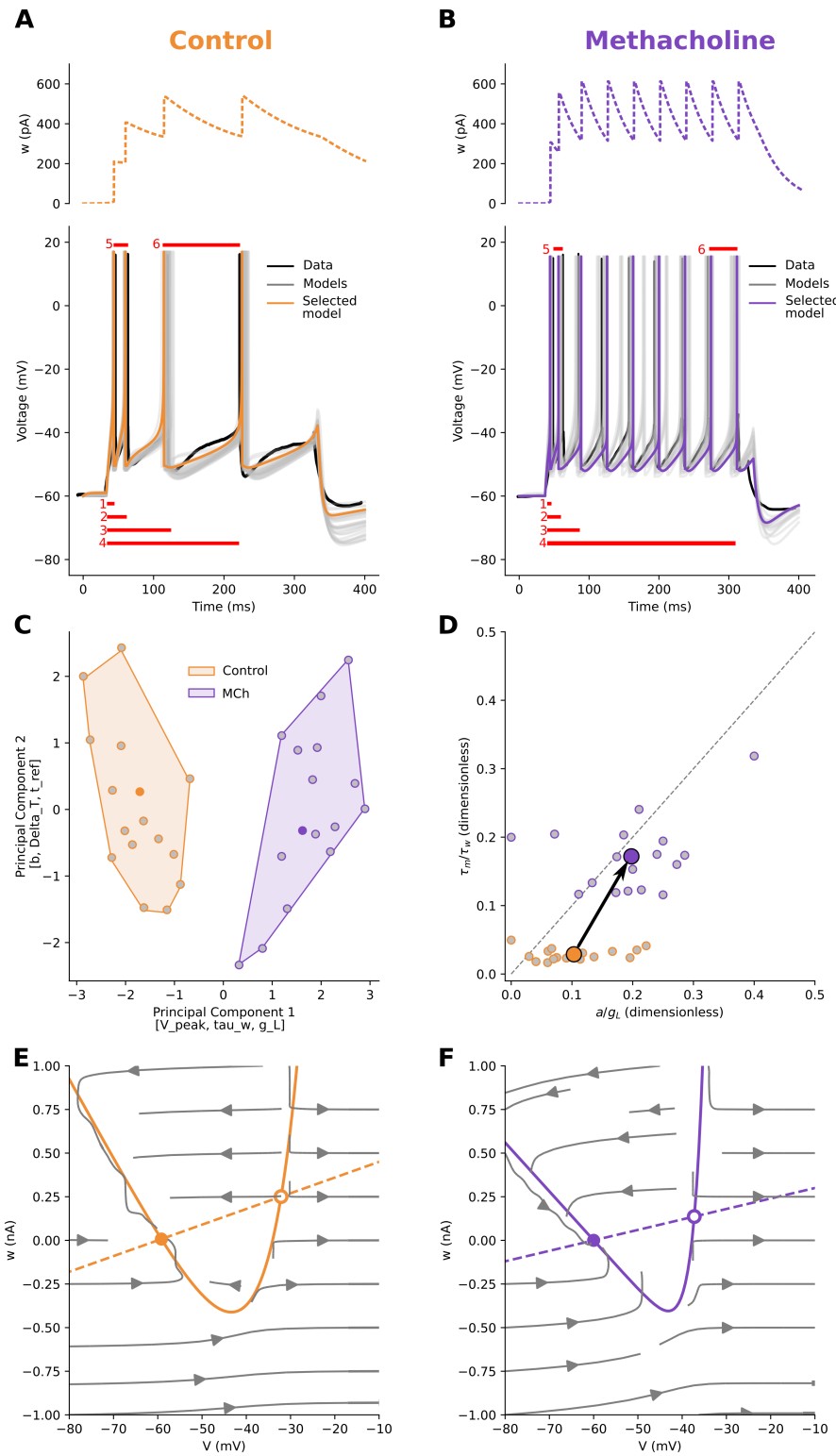

**Fig 1**. **How neuromodulators control the excitability of neurons. A:** Cortical pyramidal neuron in control condition. Electrophysiology data (black), multiple fitted AdEx models (grey), and one selected model (orange, same colour throughout the paper) are presented together (Voltage traces, solid, on bottom, adaptation current, dashed, on top). The features measured over the traces were: time to first, second, third, and last spike (red bars 1, 2, 3, 4), inverse of the first and last interspike interval (bars 5 and 6), and firing frequency. **B:** The same pyramidal neuron in Methacholine (MCh) bath application. The voltage and adaptation traces are appreciably different. The same features as in A were measured. **C:** To study the relationship

between the two conditions, we applied a dimensional reduction technique (PCA) to the parameters of fitted (grey points) and selected (centroid, same colour as in the respective panels) models. In this space, where the top three features contributing to the first principal component were [`V_peak`, `tau_w`, `g_L`] and the top contributors of the second principal component were [`b`, `Delta_T`, `t_ref`], the control and modulated clusters were distinct (Silhouette score $\sim 0.55$). **D:** xcitability landscape, where each fitted model is projected onto two biophysical ratios. The control condition (orange) is near the bottom of the excitability landscape, below the diagonal, in Class II, where the resting state is lost by a subcritical Hopf. Methacholine (violet) scales the mean up and to the right, producing stronger, faster subthreshold adaptation relative to leak. The mean moves closer to the diagonal and to switching (small circles: individual fits; filled symbols with black arrows: condition means and mean Control→Modulated displacement; dashed line: Bogdanov–Takens boundary). **E, F:** To understand the effects of neuromodulation in terms of excitability dynamics, we plotted the `V` (solid) and `w` (dashed) nullclines, with their intersection corresponding to fixed points. With respect to control (orange), MCh (violet) increases the excitability of the neuron by lowering the slope of the V-nullcline left branch, while also reducing the adaptation rate (w-nullcline slope).

We extracted and collected the key features of seven types of neurons – Pyramidal, direct and indirect Striatal projection, Dentate Granule cells, Relay, Reticular, Granule cells – from five areas of the human and rodent central nervous system – Cortex, Striatum, Hippocampus, Thalamus and Cerebellum – under the effect of five types of neuromodulators – Acetylcholine, Norepinephrine, Serotonin, Histamine, and Dopamine (Table 1).

We modelled traces using the Adaptive Exponential Integrate-and-Fire (AdEx) model [12,13]. Given feature values and parameter bounds, our search algorithm sampled parameter sets, simulated traces, extracted features, computed errors, and ranked models. The 16 best fits (lowest error) were retained (Fig 1A,B; S2 Fig) for each condition.

When focusing on a specific neuron type, in this case cortical pyramidal neurons (Fig 1), it is possible to assess the consistency of this approach and examine the insights it may provide. To test whether neuromodulation systematically shifts a neuron's parameters space, we applied principal component analysis (PCA) to the fitted models (Fig 1C). In all cases, control and modulated conditions formed distinct clusters (with a high Silhouette score, which measures how well the samples from a condition are similar to one another and distinct from those of another condition). As a consistency check, three recordings from the same human cortical neuron [15] were fitted and analysed as if they were independent; their PCA clusters overlapped (Silhouette < 0), confirming that they are in fact non-independent (S3 Fig).

The PCA (Fig 1C) identifies the parameters that separate Control and MCh clusters. In turn, the excitability landscape (Fig 1D) explains how the relationship between these parameters determines the observed behaviour. In this dataset, PC1 carries most of the variance from $g_L$, $\tau_w$, and $V_{peak}$, whereas PC2 is dominated by $b$ and $\Delta T$. Control (orange) forms a compact cluster on the left, while methacholine (MCh, a non-specific cholinergic agonist, violet) forms a distinct cluster on the right, indicating a coherent, multi-parameter shift under MCh.

Panel D projects each fit onto two biophysical ratios that determine the onset bifurcation in AdEx: $R_a = a/g_L$, $R_\tau = \tau_m/\tau_w = C_m/(g_L \tau_w)$. Because both ratios divide by $g_L$, a decrease in $g_L$ moves the points right/up, towards a switch of bifurcation class. Likewise, decreasing $\tau_w$ raises $R_\tau$. The control condition (orange) is near the bottom of the excitability landscape, with small $R_a$ and small $R_\tau$, yet below the diagonal $R_a = R_\tau$, therefore, in Class II, where the resting state is lost by a subcritical Hopf. MCh (violet) shifts the system towards the switch: both $R_a$ and $R_\tau$ increase (stronger, faster subthreshold adaptation relative to leak and membrane), and the mean moves closer to the Bogdanov–Takens line.

The Fig 1E, F showcases the insights provided by this approach for a pyramidal neuron in control condition (orange) and under the effect of MCh (violet). In the phase plane, the leak conductance $g_L$ is proportional to the slope of the left branch of the V-nullcline (Fig 1E, F, solid curves). A steeper slope on the V-nullcline and a higher $g_L$ indicate that a small change in the adaptation variable $w$ leads to a relatively small change in the membrane potential, resulting in a slower approach to the threshold. Conversely, a shallower slope of the V-nullcline implies that a small change in the adaptation variable $w$ leads to larger changes in the membrane potential, resulting in a faster approach to the firing threshold. The vertex of the V-nullcline marks the point of maximal sensitivity of the membrane potential to the adaptation current $w$. In practical terms, it corresponds to the minimum of the V-nullcline, i.e., the lowest adaptation current at which $dV/dt = 0$. The horizontal position of the vertex (in V) roughly indicates the threshold region where the neuron is

transitioning from rest to spiking. The vertical position (in $w$) reflects the minimum adaptation current needed to keep the neuron at rest at that voltage. Similarly, in the phase plane, the slope of the $w$-nullcline is proportional to the parameter $a$ (Fig 1E, F, dashed lines). A steeper slope on the $w$-nullcline indicates that a small change in the voltage variable $V$ leads to a large change in the adaptation variable, which is a subtracting term in computing the voltage variation $dV$, thus reducing the voltage change and resulting in a slower approach to the threshold. Conversely, a shallower slope of the $w$-nullcline implies that a small change in the adaptation variable $w$ results in small changes in the membrane potential, affecting its approach to the firing threshold. The parameter $b$ shifts instantaneously $w$, but it does not affect the slope of the $w$-nullcline.

Since MCh is a non-specific cholinergic agonist and mimicks the effect of Acetylcholine (ACh), by comparing the phase planes for control (orange) and modulated (violet) systems (Fig 1E, F), we can clearly understand how ACh increases the excitability of the neuron by lowering the $V$-nullcline and $w$-nullcline slopes. Indeed, ACh controls input resistance with receptors opening and/or closing non-voltage-gated $K^+$ and $Na^+$ channels. Combinations of these "leak" permeabilities, $g_L$ (in the AdEx), provide a mechanism by which neuromodulators can dynamically shift excitability regimes as their receptors are activated over time [38]. Similarly, voltage-dependent muscarinic currents (IM), $w$ (in the AdEx), have a slow component activated upon depolarisation, controlled by the parameter $a$ (in the AdEx), which is modulated by ACh [4,16]. The magnitude of spike-induced IM, controlled by parameter $b$ (in the AdEx), is also modulated by ACh [17].

### Cortical pyramidal neurons

We applied our parameter search procedure to the traces from the work of [15] on human cortical pyramidal neurons in control conditions (Fig 1A) and under the bath application of Histamine (HT, Fig 2A), Methacholine (MCh, Fig 2B, as Fig 1B), Norepinephrine (NA, Fig 2C), and Serotonin (5-HT, Fig 2D). We extracted the electrophysiological features (as in Fig 1A and B); fitted AdEx parameter sets for each condition, applied dimensionality reduction to ease their grouping (Fig 2E), and finally studied their exitability plane and dynamics (Fig 2F).

Histamine, HT (Fig 2A, and 2E dark green points, boundary, and shaded area), is the neuromodulator that impacts the least firing rate and other features of pyramidal neurons' firing. From the reduced PCA space and the phase space (Fig 2E and insets in panel F) we can see that, with respect to control conditions, HT does not alter the leak conductance ($g_L$) and the spike-triggered adaptation ($b$), but it decreases the slope of the $w$-nullcline (parameter $a$) and the adaptation time constant ($\tau_w$). The parameter $a$ bundles adaptation conductances responsible for voltage-gated $K^+$ currents (M-type currents), $Ca^{2+}$ currents, and $Ca^{2+}$-gated $K^+$ (AHP-type) currents [16]. As such, to capture the complex interactions of these currents, it can also take negative values in the AdEx model — a mathematical abstraction since conductances are not physically negative. HT decreases $Ca^{2+}$-activated $K^+$ conductances, resulting in small depolarization and increased input conductance [18–20].

Methacholine, MCh, a non-specific cholinergic agonist (Fig 2B, and 2E violet points, boundary, and shaded area) has been described above (Fig 1B). Here we just note the effects of reduction of leak conductance ($g_L$), adaptation decay time ($\tau_w$) and its unitary increase ($b$), common to all neuromodulators. As said above, this increases neuron excitability by lowering the $V$- and $w$-nullcline slopes. In contrast to other neuromodulators, MCh does not alter the spike upswing rate ($\Delta_T$) and peak ($V_{peak}$) with respect to the control condition.

Norepinephrine, NA (Fig 2C, and 2E blue points, boundary, and shaded area), lowers the leak conductance ($g_L$), the adaptation decay time ($\tau_w$) and its unitary increase ($b$). At the same time, it raises the exponential spike raise time ($\Delta_T$). These changes are reflected in the phase space (Fig 2F, insets). NA increases the neuron's excitability by lowering the slope of the left branch $V$-nullcline (solid line). The shallower $V$-nullcline slope would result in a faster approach to the firing threshold. But, at the same time, NA is the only neuromodulator that raises the slope on the $w$-nullcline (dashed line). Thus, a small change in the voltage variable $V$ leads to a large change in the adaptation variable, subtracting more from the voltage variation $dV$, reducing the voltage change and balancing the effects of a shallow $V$-nullcline. This results in

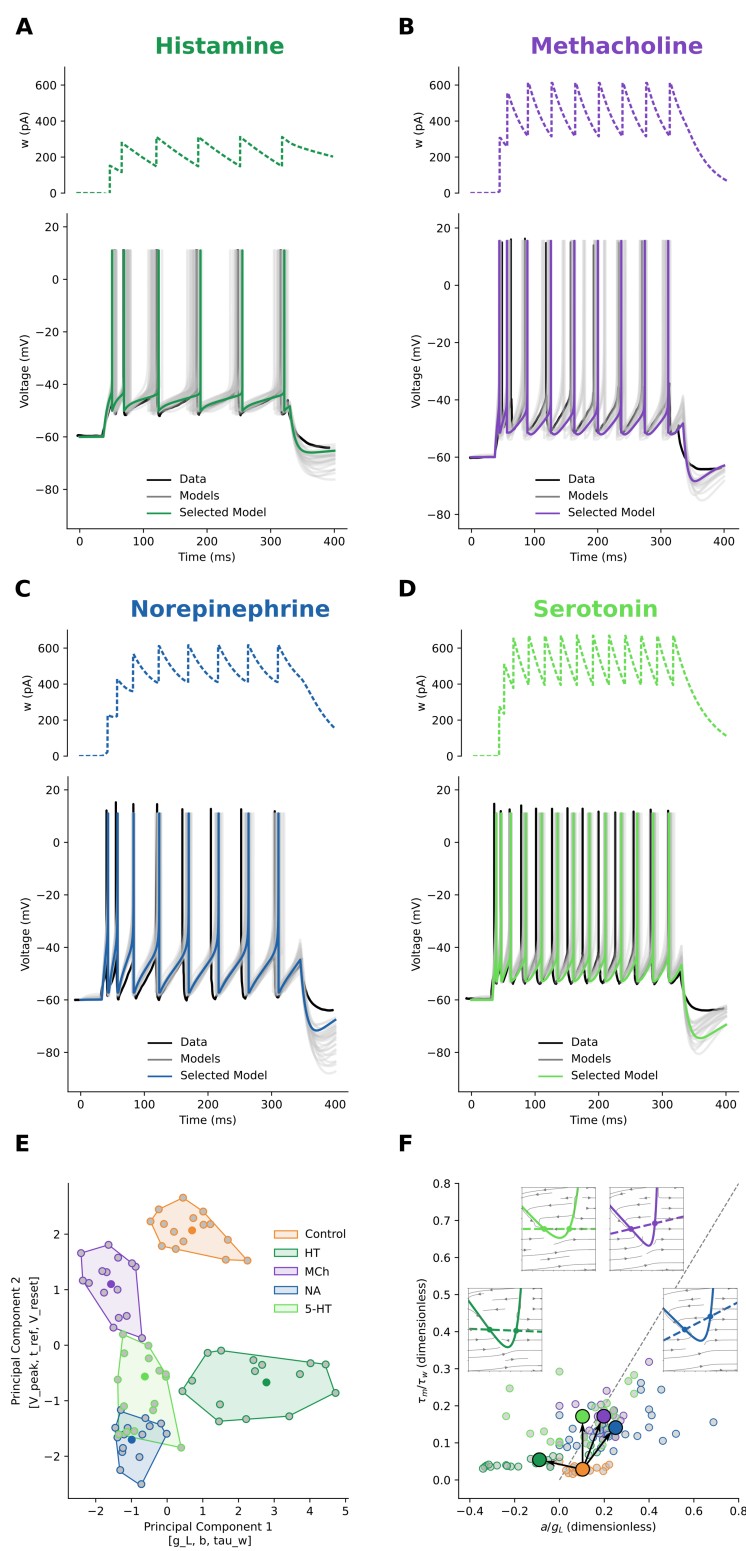

**Fig 2. Pyramidal neuron spiking behaviour is either switched or scaled**. With respect to the control condition (in Fig 1A): **A:** Histamine (HT) reduces the inter-spike-interval (ISI). **B:** Methacholine (MCh) reduces more ISI and time-to-first-spike (TFS). **C:** Norepinephrine (NA) has similar effects of reduced ISI and TFS. **D:** Serotonin (5-HT) has the largest impact on both ISI and TFS. In all panels from A to D, top: model adaptation variable *w*, bottom: Voltage, with the original scraped data (black), fitted models (grey), and the selected models (same colours throughout the paper). **E:** To study

the relationship between the conditions, we applied a dimensionality reduction technique (PCA) to the parameters of fitted (grey points) and selected (centroid, same colour as in the respective panels) models. In this space, the control (orange) and modulated clusters were distinct (Silhouette score $\sim 0.4$). The top three features contributing to the first principal component were [`g_L`, `b`, `tau_w`] and the top contributors of the second principal component were [`V_peak`, `t_ref`, `V_reset`]. HT cluster presents the largest changes along the first principal component, while 5-HT and MCh have the largest variation along the secondary principal component. **F:** In the excitability landscape, modulators that shift rightward in parameter space map to leftward displacements in the excitability landscape, yielding the observed mirror: ACh/HT/5-HT switch toward the SNIC subspace, whereas NA scales reinforcing type-II (sub-Hopf) excitability. Small circles: individual fits; filled symbols with black arrows: condition means and mean Control→Modulated displacement. Dashed line: Bogdanov–Takens boundary. Insets show the $V$ (solid) and $w$ (dashed) nullclines for each neuromodulators. While HT (dark green), MCh (violet), and 5-HT (light green) reduce the adaptation rate (lowering the $w$-nullcline slope), NA (blue) increases adaptation but also further raises the vertex of the $V$-nullcline.

a small relative increase in spiking frequency (control $\sim$ 13Hz vs. NA 26 Hz). Indeed, as reported by [15] and [21], the activation of $\beta$-adrenergic receptors in pyramidal neurons results in an enhanced excitability, mediated by a slow $Ca^{2+}$-activated $K^+$ current.

Serotonin, 5-HT (Fig 2D, and 2E light green points, boundary, and shaded area), more than MCh but less than NA, lowers the leak conductance ($g_L$), the adaptation decay time ($\tau_w$) and its unitary increase ($b$), and raises the exponential spike raise time ($\Delta_T$). In the phase space (Fig 2F, insets), 5-HT does not alter much the slope of the $V$-nullcline (solid line) but increases the excitability of the neuron by flattening the slope on the $w$-nullcline (dashed line, values of $a$ are close to 0). Thus, changes in the voltage variable $V$ have little effect on the adaptation variable, which in turn affects the voltage variation $dV$ less, resulting in no firing adaptation. Both HT and 5-HT cause a raise in firing rate through a shift in excitability (a type-I SNIC bifurcation), but for quite orthogonal reasons. HT shifts the excitability by reducing IM conductances without altering leak conductances. McCormick [19] reported that 5-HT has an excitatory effect suppressing IM currents. Indeed, recently it has been found [22,23] that the activation of 5-HT$_{2A}$ receptors, affects IM currents by reducing their duration. Indeed this is well visualized in our excitability landscape.

The parameter space (Fig 2E) correlates with the excitability landscape (Fig 2F). Panel E shows the PCA of the fitted AdEx parameters. In this dataset, PC1 carries most of the variance from $g_L$, $b$, and $\tau_w$. Because both excitability coordinates (in 2F) are inversely proportional to $g_L$ (and $\tau_m/\tau_w$ also to $\tau_w$), modulators that shift rightward in parameter space map to left displacements in the excitability landscape, yielding the observed mirror correlations: ACh, HT, and 5HT switch toward the type-I (SNIC bifurcation), with gain of arbitrarily low rates. HT also increases first-spike latencies. Norepinephrine scales up, reinforcing type-II (sub-Hopf) excitability, with short onset latencies and a finite minimal firing rate.

## Striatal projection neurons

The striatum is the largest nucleus and main input stage of the Basal Ganglia. It receives input from the cerebral cortex, thalamus and midbrain. Around 95% of the neurons in the rodent striatum are striatal projection neurons (SPNs), and the remaining 5% are interneurons [24]. SPNs are approximately equally divided into direct striatal projection neurons (dSPNs) and indirect striatal projection neurons (iSPNs). The dSPNs express D1 dopamine receptors, which have a high density in the striatum and when activated by Dopamine (DA) increase the neuron excitability. On the contrary, the iSPNs express D2 dopamine receptors, which decrease the neuron excitability when activated [25].

We applied our parameter search procedure to the available experimental data on the effects of DA on SPNs from [26] (their Fig 4C, control and after bath application of DA in mice animal model) to capture the dichotomous effects of DA on AdEx neuronal excitability (Fig 3). In addition, we applied the same procedure to the experimental data on the effects of Acetylcholine (ACh) from [27] (their Fig 8A and B, in rat animal model).

We grouped the six models into two conditions: control (Fig 3A-C, different shades of orange for dSPN, iSPN, and SPN) and under the influence of DA and ACh (Fig 3E-F, DA-modulated dSPN in blue, DA-modulated iSPN in cyan, and

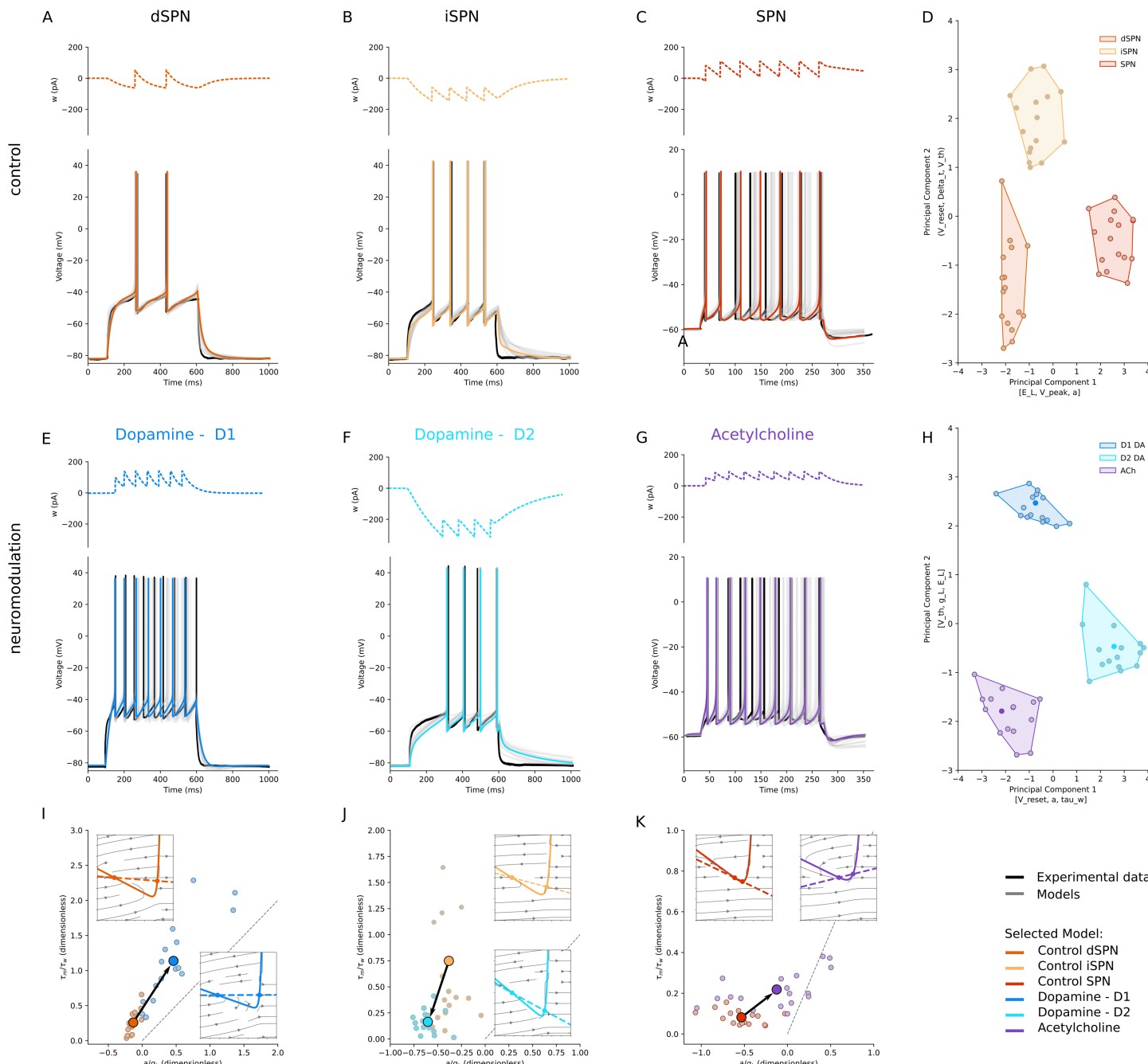

**Fig 3**. **Striatal neurons sharpen and reverse their differences.** D1-receptor expressing SPN (dSPN) and D2-receptor expressing SPN (iSPN) behave differently under the effects of Dopamine (DA). In control conditions, dSPNs **(A)** have low, non-adapting spiking rates, with long latency to spike, and iSPNs **(B)** have a similar behaviour **C:** Regardless of their receptor expression, more depolarised conditions only increase the firing rate of SPNs without increasing spiking adaptation. **D:** In the PCA-reduced parameter space, the behaviour of dSPN (dark orange) and iSPN (light orange) is distinguished by approach to threshold and after-spike reset. A different holding voltage, without considering dopamine receptor expression (red), shifts the parameters along the Principal Component 1 axes [`E_L`, `V_peak`, a]. The Silhouette score is $\sim 0.68$. **E:** In DA bath conditions, D1-expressing SPNs lose the latency to first spike and triple their spiking rate. **F:** Conversely, in the same condition, D2-expressing SPNs increase their latency and not their firing. **G:** Acetylcholine (ACh) increases the firing rate of SPNs and introduces spike adaptation. **H:** In the PCA space, DA increases the difference between dSPNs and iSPNs (Silhouette score in control condition is $\sim 0.59$, while under the effect of DA is $\sim 0.66$) and in general improves the grouping of the clusters (Silhouette score goes from $\sim 0.68$ to $\sim 0.74$). **I:** For dSPN, D1 dopamine, the displacement is right–upward ($a/g_L \uparrow$, $\tau_m/\tau_w \uparrow$) and the mean scales along the BT line. Insets show the $V$ (solid) and $w$ (dashed) nullclines, as in J and K. Phase planes show a flatter $w$-nullcline and approach to the left knee

of the $V$-nullcline, predicting continuous $f$–$I$ onset and longer near-rheobase latencies. **J**: For iSPN, D2 dopamine, the displacement is left–downward ($a/g_L \downarrow$, $\tau_m/\tau_w \downarrow$) **K**: For a generic SPN, in ACh condition, the shift is mainly rightward ($a/g_L \uparrow$) with a modest vertical component; the mean moves close to switching.

Fig 3G, ACh-modulated SPN in violet), and we performed PCA on the control and modulated parameter sets, as shown in Fig 3D and H, respectively. In both cases, the models form distinct clusters, but neuromodulation increases the separation between them, as indicated by a rise in the Silhouette score from $\sim 0.68$ to $\sim 0.74$. Finally, we investigated the effect of DA and ACh on the corresponding control models by examining their dynamics in the phase plane and excitability landscape (Fig 3I-K).

In dSPNs (Fig 3E), DA has an excitatory effect. It decreases the leak conductance $g_L$, resulting in a shallower slope of the $V$-nullcline (solid lines in insets in Fig 3I), implying that a small change in the adaptation variable $w$ leads to larger changes in the membrane potential, resulting in a faster approach to the firing threshold. It also makes the slope of the $w$-nullcline shallower (dashed lines in insets in Fig 3I), such that a change in the adaptation variable $w$ leads to smaller changes in the membrane potential, affecting less its approach to the firing threshold. In addition, a decrease in the spike threshold ($V_{th}$) alongside a reduction in the adaptation parameters $b$ and $\tau_w$, results in a slower approach to threshold. Indeed, by reducing the leak conductance ($g_L$) and enhancing $Na^+$ conductance, dopamine scales the neuron's excitability within a Class I (SNIC) regime[26,28].

In iSPNs (Fig 3F), DA has an inhibitory effect. It increases the leak conductance $g_L$, reflecting an increase in $K^+$ conductance, steepening the slope of the left branch of the $V$-nullcline (solid lines in insets in Fig 3J), resulting in a slower approach to the threshold. It also steepens the slope on the $w$-nullcline (dashed lines in insets in Fig 3J) such that a small change in the voltage variable $V$ leads to a large change in the adaptation variable, which is a subtracting term in computing the voltage variation $dV$, thus reducing the voltage change and resulting in a slower approach to threshold. Moreover, the increase in the adaptation parameters $b$ and $\tau_w$ and decrease of $a$ underscores a heightened delay in the first spike and adaptation response.

In summary, DA can modulate $Na^+$, $K^+$, and $Ca^{2+}$ channels. An increase or decrease in $g_L$ might reflect an increase or decrease in $Na^+$ and $K^+$ conductance, leading to hyperpolarization or depolarization. Changes in the adaptation parameters ($a$, $b$, and $\tau_w$) could suggest alterations in $Ca^{2+}$ channel activity affecting neuron's firing patterns. It is important to stress that the parameters $g_L$, $a$, $b$, $\tau_w$, $V_{reset}$ are modulated in the opposite directions depending on the DA receptor type (see S1 Appendix).

On SPNs, without considering the expression of DA receptors, ACh increases excitability compared to control (Fig 3C and G). In line with the expected influence of ACh, the AdEx models exhibit a decrease in the leak conductance $g_L$, representative of enhanced $Na^+$ channel permeability, and $V_{th}$, indicative of a facilitated response to incoming stimuli.

The parameter space (Fig 3D and H) correlates with the excitability landscape (Fig 3I-K). Both for dSPNs and iSPN under DA (panel I and J, respectively), the mean scales without crossing the diagonal, keeping a Class I SNIC. However, while for dSPNs the displacement is right-upward (larger $a/g_L$ and $\tau_m/\tau_w$), for iSPNs is left-downward (smaller $a/g_L$ and $\tau_m/\tau_w$), in perfect agreement with the known dichotomous effects of DA. ACh in SPNs (panel K) brings the mean close to switching (larger $a/g_L$) with a modest vertical component. The insets show a shallower slope of the $V$-nullcine and flattened $w$-nullcline which result in increased excitability.

## Hippocampal neurons

In the hippocampal formation, the Dentate Gyrus (DG) occupies an important input position, receiving from the entorhinal cortex and projecting to the first portion of the Cornus Ammonis (CA1). The principal neurons of the DG are the granule cells.

To characterize dentate granule cells, we applied our parameter search procedure to electrophysiological traces from the works of [29] (their Fig 5, control vs bath application of oxo-M, a muscarinic agonist, mimicking ACh conditions, in mice), [30] (their Fig 3, HT condition, in rats), and [31] (their Fig 2, bath application of Isoproterenol, a $\beta$-adrenergic agonist, mimicking NA condition, in rats). We report the fitting for the control condition (Fig 4A) and under modulation of ACh (Fig 4B), HT (Fig 4C), NA (Fig 4D). As before, we measured the electrophysiological features; we obtained AdEx parameter sets for each condition through fitting, and then we applied dimensionality reduction to ease their grouping (Fig 4E). Once we found these groups, we studied their excitability landscape and dynamics (Fig 4F).

In the original work from [29], excitatory input currents of 60, 100, 140 pA were used in both the control and ACh conditions. We fitted our model to all the values in the control condition, and the 140 pA data because of its higher quality and consistency. The authors also reported the use of a holding current. However, they did not specify its intensity. For our fitting, we have assumed –300 pA for the control and –200 pA in the condition of muscarinic agonist oxo-M bath application. This is because ACh increases the input resistance, and less current is usually needed to hold the neurons at the desired voltage (–75 mV, here and in the original article).

In control conditions (Fig 4A, orange point and shaded area), dentate granule cells strongly adapt their firing. In this condition, the left branch of the $V$-nullcline (solid line in inset in Fig 4F) is high, with high leak currents, and the $w$-nullcline (dashed line in inset in Fig 4F) has a high positive slope, providing a powerful adaptation current. Our fitting also places high values for the $\tau_w$, stretching the effects of even low $b$ unitary increases to $w$.

Compared to the control condition, the bath application of muscarinic agonist oxo-M (Fig 4B, violet points and shaded area in E) increases the excitability of dentate granule cells by lowering the left branch of the $V$-nullcline, and bringing the $w$-nullcline to negative slope ($a$). The effects of ACh are similar on granule cells and cortical pyramidal neurons. Indeed, as for pyramidal neurons, ACh controls non-voltage-gated $K^+$ and $Na^+$ channels [29,32], and it also slows voltage-dependent muscarinic-dependent $Ca^{2+}$-activated adaptation currents [29,62]. By reducing adaptation and effectively lowering $a$ while maintaining leak conductance, ACh shifts neuronal excitability toward the Class I (SNIC) domain, promoting firing frequency increases and enhanced responsiveness to depolarizing inputs.

HT application (Fig 4C, green points in E, and green lines in F) does not lead granule cells to an increase in spiking frequency as strong as ACh and NA. However, it shifts the parameter space of the dentate granule cells in a different direction. It lowers the left branch of the $V$-nullcline (solid line in inset in Fig 4F), and therefore its leak currents. But the $w$-nullcline (dashed line in inset in Fig 4F) has a positive slope close to the control condition, which provides a powerful adaptation current, balanced by low values for $\tau_w$, which shortens the effects of $b$ unitary increase to $w$. At the same time, HT raises the reset voltage ($V_{reset}$) and the spike upswing rate ($\Delta_T$), leading to smaller depolarisation and a slower approach to threshold. Indeed, [30] showed that HT causes, through specific $H_2$-receptors, a block of the long-lasting component of spike after-hyperpolarisation (AHP) and a reduction of firing adaptation. In terms of excitability class, this combination of higher reset potential and smaller $\Delta_T$ shifts the neuron toward the Class I domain, characterized by an SNIC bifurcation and graded firing frequencies. In addition, HT selectively blocks the late long-lasting $Ca^{2+}$-dependent outward tail current without any reduction of inward current, as captured by the model $V$- and $w$-nullclines.

NA has a similar effect to ACh on dentate granule cells (Fig 4D, blue point and shaded area in E). NA increases the excitability of dentate granule cells by lowering even more than ACh the left branch of the $V$-nullcline (solid line in inset in Fig 4F), and bringing the $w$-nullcline (dashed line in inset in Fig 4F) to 0 (or negative) slopes. The similarity between the two neuromodulators is clear in Fig 4E, where the two parameter clusters (violet for ACh and blue for NA) are distinct from one another but close compared to control and HT conditions. Indeed, [31] showed that NA reduces dentate granule cell spike adaptation through $\beta$-receptors blockade of $Ca^{2+}$-dependent $K^+$ conductance. By diminishing adaptation and thus lowering the effective coupling parameter $a$, NA shifts the neuron's dynamics toward the Class I (SNIC) regime, favoring continuous frequency modulation and enhanced excitability.

In general, the parameter space (Fig 4E) correlates with the excitability landscape (Fig 4F). ACh, NA, and HT switch neurons from excitability of Class II to Class I. However, ACh and NA cause little change for the ratio $\tau_m/\tau_w$, reducing

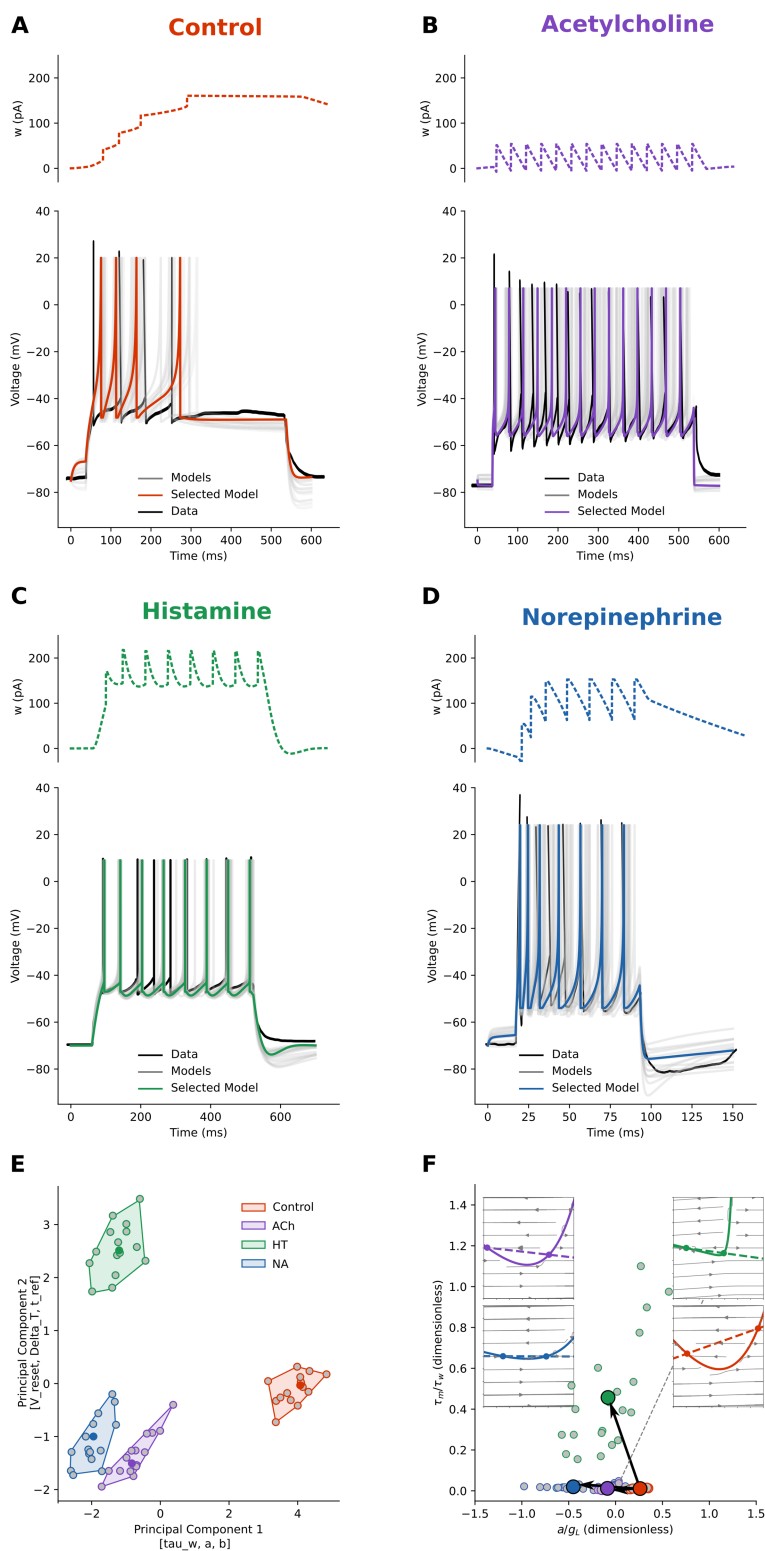

**Fig 4**. **Dentate granule cells have polarised responses to neuromodulators switch spiking behaviour under neuromodulation. A:** In control conditions, DGCs strongly adapt their firing. **B:** Acetylcholine (ACh) increasingly reduces inter-spike-interval (ISI) and abolishes spike adaptation. **C:** Histamine (HT) reduces spike adaptation but leaves ISI close to control conditions. **D:** Norepinephrine (NA) has effects of reduced spike adaptation and ISI similar to ACh. **E:** In the PCA-reduced parameter space, the control (red) and modulated clusters were distinct (Silhouette score ∼ 0.6). HT cluster

(light green) presents the largest variation along both Principal Component axes, and NA and ACh have similar variations. The parameters most influencing PC1 are related to subthreshold and adaptation properties [`Delta_T`, `b`, `tau_w`], while PC2 is primarily shaped by spiking threshold and reset dynamics [`V_th`, `V_peak`, `V_reset`]. **F:** All neuromodulators caused a switch in the excitability landscape, shifting the ratio $a/g_L$ downward and moving the cells from Class II to Class I excitability. Small circles: individual fits; filled symbols with black arrows: condition means and mean Control→Modulated displacement. Dashed line: Bogdanov–Takens boundary. In the inset phase spaces, we have the $V$ (solid) and $w$ (dashed) null-clines, with their intersection representing fixed points. With respect to control (orange), all neuromodulators, except ACh, increase the excitability of the neuron by lowering the slope of the $V$-nullcline left branch and raising the vertical position (in $w$) of the $V$-nullcline vertex. While HT (dark green) and NA (blue) reduce the adaptation rate (lowering the $w$-nullcline slope), ACh increasingly removes adaptation (captured by the negative $w$ slope) and also lowers the vertex of the $V$-nullcline.

firing-rate accommodation with respect to the control condition, without altering onset type. HT causes a larger effect for the ratio $\tau_m/\tau_w$, introducing longer first-spike latencies and abolishing firing-rate accommodation.

## Thalamocortical projecting neurons

The thalamus is a central structure located in the forebrain above the midbrain. It acts as a central hub, with nerve fibres — the thalamocortical radiation — connecting it to all parts of the cerebral cortex. Nearly all thalamic neurons (except those in the reticular nucleus) project to the cortex, and every cortical region sends feedback to the thalamus, but collateral connections reach the basal ganglia, hippocampus, colliculus, and cerebellum. It relays and processes sensory and motor signals and plays key roles in consciousness, sleep, and alertness [63].

To characterise thalamocortical projecting neurons (TPNs), we applied our parameter search procedure to electrophysiological traces from several works. In the absence of neuromodulators, the TPNs of all nuclei display a bursting activity mode. For our fitting of this mode, we used mouse Centro-Lateral neuron data from the work of [33], for their clarity, but analogous voltage responses can be seen in the works of McCormick [34,35,38]. In all thalamic nuclei, ACh provided by brainstem nuclei [36,37] causes the switch to tonic firing mode [48]. For our fitting, we used the traces from [38] (their Fig 4A, right). Similar effects are induced by Orexin-B (Ox), a drug targeting channels also targeted by Dopamine, and NA. We used [33] for Ox (their Fig 5, control vs Orexin-B bath conditions) and [38] for NA (their Fig 2B).

In our Fig 5, we report the fitting for the control (5A) and ACh (5B), Orexin-B (5C), NA (5D) conditions. As before, we measured the electrophysiological features, we obtained AdEx parameter sets for each condition through fitting, and then we applied dimensionality reduction to ease their grouping (Fig 5E). Once we found these groups, we studied their excitability landscape and dynamics (Fig 5F).

In the control condition, where the $E_L$ is low (around −70 mV, Fig 5A, orange points and shaded area in E), TPNs strongly adapt their firing. The low $E_L$ coupled with relatively high $g_L$ and adaptation $a$ give a slow approach to threshold. The left branch of the $V$-nullcline (solid line in inset in Fig 5F) is high, with high leak currents, and the $w$-nullcline (dashed line in inset in Fig 5F) has a relatively high positive slope, providing a powerful adaptation current. In fact, the absence of ACh in the thalamus reduces TPNs excitability due to a muscarinic receptor-mediated increase in non-voltage dependent $K^+$ conductance that lowers the resting membrane potential [38]. When stimulated in this condition, TPNs generate a $Ca^{2+}$ spike that triggers a burst of fast $Na^+/K^+$ action potentials [38]. Such burst-prone dynamics indicate a shift toward the Class I excitability domain, with graded frequency modulation. The bath application of ACh (Fig 5B, violet points and shaded area in E, and violet lines in F) reduces the adaptation of TPNs and increases their excitability due to the reverse mechanism described above, muscarinic receptor-mediated decrease in $K^+$ conductance [38]. In these conditions, TPNs fire tonically, with ACh depolarising the neuron out of the voltage range in which the low threshold $Ca^{2+}$ current is active (Fig 4A in [38]). This transition corresponds to a shift toward the Class I (SNIC) excitability regime, where the suppression of burst-generating $Ca^{2+}$ currents and reduction of adaptation enable smooth, continuous frequency modulation. The AdEx model does not contain a term explicitly representing $Ca^{2+}$ currents, but the sub-threshold adaptation $a$, the low $E_L$

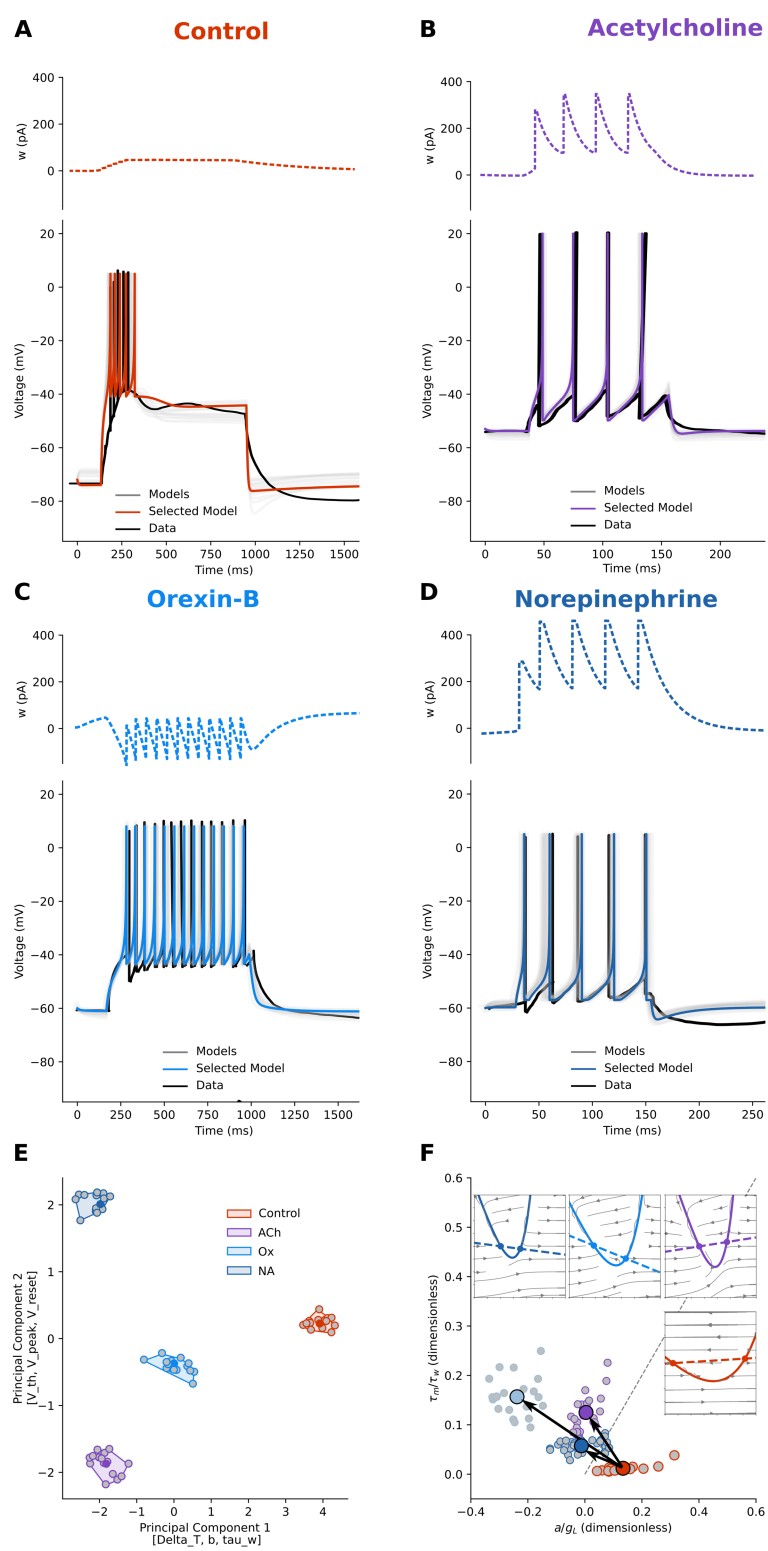

**Fig 5**. **Thalamocortical projecting neurons under neuromodulation switch from Class II to Class I. A:** In control conditions, TPNs strongly adapt their firing. A spiking behaviour known as "bursting". All neuromodulators make the TPNs switch to a regular spiking behaviour known as "tonic" firing. **B:** Acetylcholine (ACh) increases the inter-spike-interval (ISI) and abolishes spike adaptation. **C:** Orexin-B (Ox) too abolishes spike adaptation and introduces latency for the time to first spike. **D:** Norepinephrine (NA) also reduces spike adaptation and has ISI similar to ACh. **E:** In the PCA-reduced

parameter space, the control (red) and modulated clusters were clearly separated (Silhouette score $\sim 0.86$). NA and ACh have similar variation along Principal Component 1 (whose top influencing parameters are [`Delta_T`, `b`, `tau_w`] but differ along the Principal Component 2 (which is mainly influenced by [`V_th`, `V_peak`, `V_reset`]). **F:** All neuromodulators caused a switch in the excitability landscape from Class II to Class I excitability. Therefore, in the phase space, we can see that TPNs' response to all neuromodulators is a narrower $V$-nullcline (solid). This corresponds to a slower approach to threshold, a less excitable system, notwithstanding the raised $E_L$ (a known effect of these neuromodulators on TPNs). Indeed, note that in the different temporal scales, all firing rates are roughly equal ($\sim 4$ sp/s). The slopes of the $w$-nullcline for the different conditions explain various spiking behaviours. In control (red), the positive $w$ slope brings strong adaptation ("bursting"). In ACh condition (violet), the even steeper $w$-nullcline is balanced by the narrower $V$-nullcline, resulting in a slower approach to threshold, which allows the decay of $w$ and the absence of adaptation ("tonic"). In Ox conditions (light blue), the (relatively strong) negative $w$-nullcline captures the latency and absence of adaptation. In NA conditions (dark blue), the negative $w$-nullcline balances the higher $V$-nullcline vertex, which allows higher adaptation currents (panel D, top).

coupled with relatively high $g_L$, and non-zero adaptation $a$ all concur in a slow approach to threshold. In the model, this is captured by a narrow $V$-nullcline.

A similar picture holds for the other neuromodulators. The application of Orexin-B (Fig 5C, data trace in black, light blue points, shaded area in E) is characterised by latency to the first spike and regular ("tonic") firing. This is due to a decrease in the leak potassium current [33]. By enhancing membrane excitability, Orexin-B shifts TPNs toward the Class I (SNIC) domain, promoting graded frequency–current relationships and sustained tonic spiking. The AdEx model does not contain a term representing the influence of Orexin-B on calcium currents, but the relatively low $g_L$ and negative sub-threshold adaptation $a$ concur in the first slow approach to threshold. After the first spike, when the $b$ term is added to $w$, the narrow $V$-nullcline (solid line in inset in Fig 5F) balances the adaptation term by slowing the dynamics and allowing the decay of $w$, contributing to the tonic firing mode.

Also the application of NA (Fig 5D, data trace in black, dark blue points, shaded area in E) shifts the TPNs from the bursting to tonic mode of activity. As described by [38], this is due to a $\alpha_1$-adrenoceptor-coupled decrease in $K^+$ conductance (their Fig 2B and C). The resulting slow depolarisation raises the input resistance and membrane time constants, and weakening adaptation. These combined effects move the system toward a Class I (SNIC) excitability regime, where neurons exhibit continuous frequency–current relationships and graded firing. In the AdEx model, the narrow $V$-nullcline (solid line in inset in Fig 5F) balances the adaptation term by slowing the dynamics and allowing the decay of $w$, effectively mimicking the increase in input resistance and time constants, thus contributing to the tonic firing mode.

In parameter space (5E), the three neuromodulators form clusters well separated from control along PC1 - dominated by $[\Delta_T, b, \tau_w]$ - and PC2, dominated by $[V_{th}, V_{peak}, V_{rest}]$. Projecting the same fits onto the excitability landscape (5F) reveals a common motif: all modulators switch the means upward by increasing $R_\tau = \tau_m/\tau_w = C_m/(g_L \tau_w)$ (i.e. faster adaptation relative to the membrane), and decreasing $R_a = a/g_L$ (a weaker brake relative to the leak). The mean trajectories crosses the Bogdanov–Takens diagonal boundary, indicating a Class II to Class I transition (subcritical Hopf to SNIC). The phase-plane insets in 5F depict the mechanism: under modulation the fixed point moves toward the left knee of the $V$-nullcline, predicting continuous $f$-$I$ onset with arbitrarily low rates, longer near-threshold latencies - all consistent with the voltage traces in panels A-D.

### Thalamic reticular neurons

The thalamic reticular nucleus encircles the thalamus laterally and dorsally. Reticular cells are GABAergic and integrate cortical [39], brainstem and mesencephalic inputs [40]. Reticular cells project outputs directly, and exclusively, to the thalamus [59,60].

To characterise thalamic reticular neurons (TRNs), we applied our parameter search procedure to electrophysiological traces from the works of [41]. However, they applied orthodromic synaptic stimuli to the neurons instead of current injections (used in all other datasets and in our modelling framework). This causes a slower decay of the membrane, which is the effect we see when comparing the models to the data (Fig 6A and B).

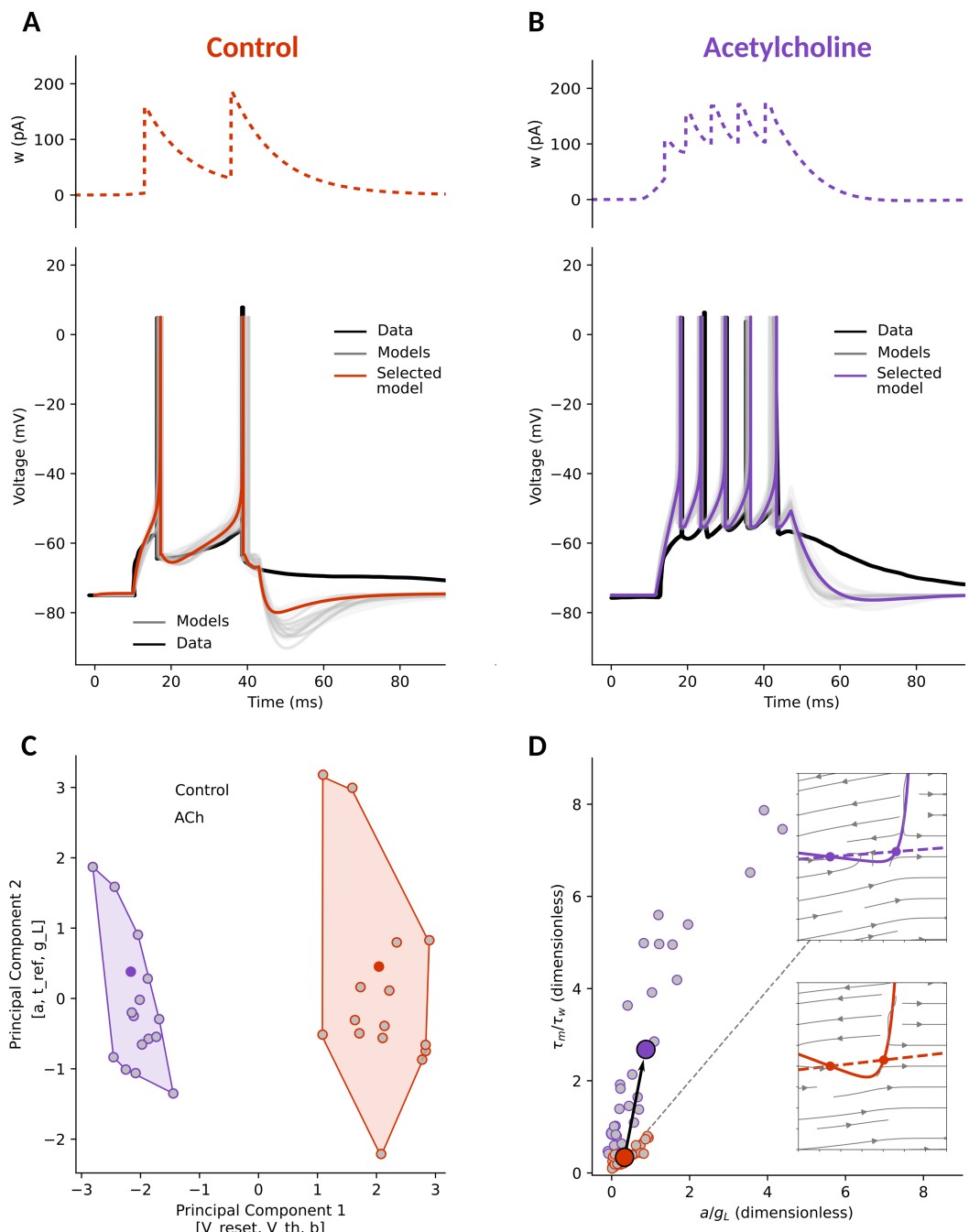

**Fig 6. Acetylcholine scales tonic firing of thalamic reticular neurons. A**: In control conditions, voltage responses to the same protocol show strong spike-frequency adaptation and intermittent bursting. **B** (ACh) The same cell fires tonically with a depolarized baseline and a shorter onset at the driven steps; adaptation is largely suppressed. **C**: The principal–component analysis of the fitted AdEx parameters separates the two ensembles (Silhouette ≈ 0.67). The displacement occurs mainly along PC1, which is dominated by the reset/threshold/leak block [`V_reset`, `V_th`, `b`], with PC2 loading primarily on subthreshold adaptation and timing terms [`a`, `t_ref`, `g_L`]. **D**: In the excitability landscape and phase planes, the mean shifts up–right from control to ACh and scales above the $R_a = R_\tau$ diagonal (Bogdanov–Takens boundary), indicating a permanence into Class I excitability. Small circles: individual fits; filled symbols with black arrows: condition means and mean Control→Modulated displacement. Dashed line: Bogdanov–Takens boundary. The phase-plane insets illustrate the underlying geometry: with ACh the the left knee of the *V*-nullcline and the *w*-nullcline become shallower, enlarging the threshold region, consistent with tonic spiking.

In control conditions (Fig 6A, orange point and shaded area in C), reticulate neurons adapt their firing. In this condition, the left branch of the *V*-nullcline (solid line in inset in Fig 6D) is high, with high leak currents, and the *w*-nullcline (dashed line in inset in Fig 6D) has a high positive slope, providing a powerful adaptation current. From [41] we know that the absence of ACh in the reticular thalamus reduces reticular neurons excitability due to a low muscarinic receptor-mediated $K^+$ conductance. In these conditions, thalamic reticular neurons fire in single spike mode [41](their Fig 3A).

Compared to the control condition, the bath application of ACh in the reticular thalamus (Fig 6B, violet points and shaded area in C) causes a membrane depolarization by lowering the left branch of the *V*-nullcline (solid line in inset in Fig 6D), and reduces spike adaptation by also lowering the slope of *w*-nullcline (dashed line in inset in Fig 6D). The effects of ACh are presumably mediated through an increase in $K^+$ conductance [41] (their Fig 3A), which diminishes the leak-driven hyperpolarizing current and alters the relative geometry of the nullclines. This rebalancing of depolarizing and adaptive currents effectively shifts the excitability of reticular neurons toward the Class I (SNIC) domain, producing smoother, continuously graded firing responses.

Also for Reticular cells, the correlation between the parameters space and the excitability landscape is direct. In the control condition, points lie near the origin of the excitability plane (small $R_a = R_\tau$) and across the diagonal. Under ACh, the mean scales up-and-right: both $a/g_L$ and $\tau_m/\tau_w$ increase. The ensemble mean becomes markedly Class I, predicting a SNIC onset mechanism. Functionally, this implies gain of arbitrarily low firing rates and longer near-rheobase latencies (continuous f–I onset). The phase-plane insets illustrate that ACh increases the excitability of the neuron by lowering both the left branch of the *V*-nullcline and the slope of the *w*-nullcline.

Considering together the thalamic relay and reticular neurons results, an interesting network property emerges. In the absence of ACh modulation, thalamocortical neurons exhibit brief spike bursts in response to stimuli. This is matched with the relatively modest single-spike responses from reticular neurons, potentially enabling stimuli to propagate further within the network. With the introduction of ACh, thalamocortical neurons show heightened tonic single-spike firing in response to stimuli. This augmented firing is mirrored by a corresponding increase in firing among reticular neurons. These findings tentatively propose that while neuromodulation reshapes the intrinsic properties and behaviours of individual cells, it may also trigger compensatory adaptations within the broader network.

## Cerebellar neurons

In the cerebellar cortex, Granule cells (GCs) are glutamatergic neurons, essential for controlling motor activity by directly exciting deep cerebellar nuclei projecting to brainstem premotor areas [64,65].

To characterize GCs, we applied our parameter search procedure to electrophysiological traces from the works of [42] for control and 5-HT conditions (their Fig 1F); and [43] for ACh condition (their Fig 6C). In the latter, Muscarine was bath applied, however, since Muscarine is a selective agonist of ACh it provides a good approximation of the modulatory effects mediated by ACh receptors. We report the fitting for the control condition (Fig 7A) and under the bath application of Muscarine (Fig 7B), Serotonin (Fig 7C). As before, we measured the electrophysiological features; we obtained AdEx parameter sets for each condition through fitting, and then we applied dimensionality reduction to ease their grouping (Fig 7D).

In control conditions (Fig 7A, orange point and shaded area in D), Granule cells present no spiking adaptation. In this condition, the left branch of the *V*-nullcline (solid line in inset in Fig 7E) is shallow, with low leak currents, and the *w*-nullcline (dashed line in inset in Fig 7E) has a negative slope, providing little adaptation current.

Compared to the control condition, the bath application of ACh (Fig 7B, violet points and shaded area in D) causes a membrane depolarisation by lowering the left branch of the *V*-nullcline (solid line in inset in Fig 7E), and shifts the *V*-nullcline vertex towards more depolarised values. In addition, it causes a progressive decrease in the interspike intervals. The effects of ACh are presumably mediated through an increase in $K^+$ conductance and channel specificity [43], which reduce the leak-driven stabilising current and reshape the balance between depolarising and adaptive

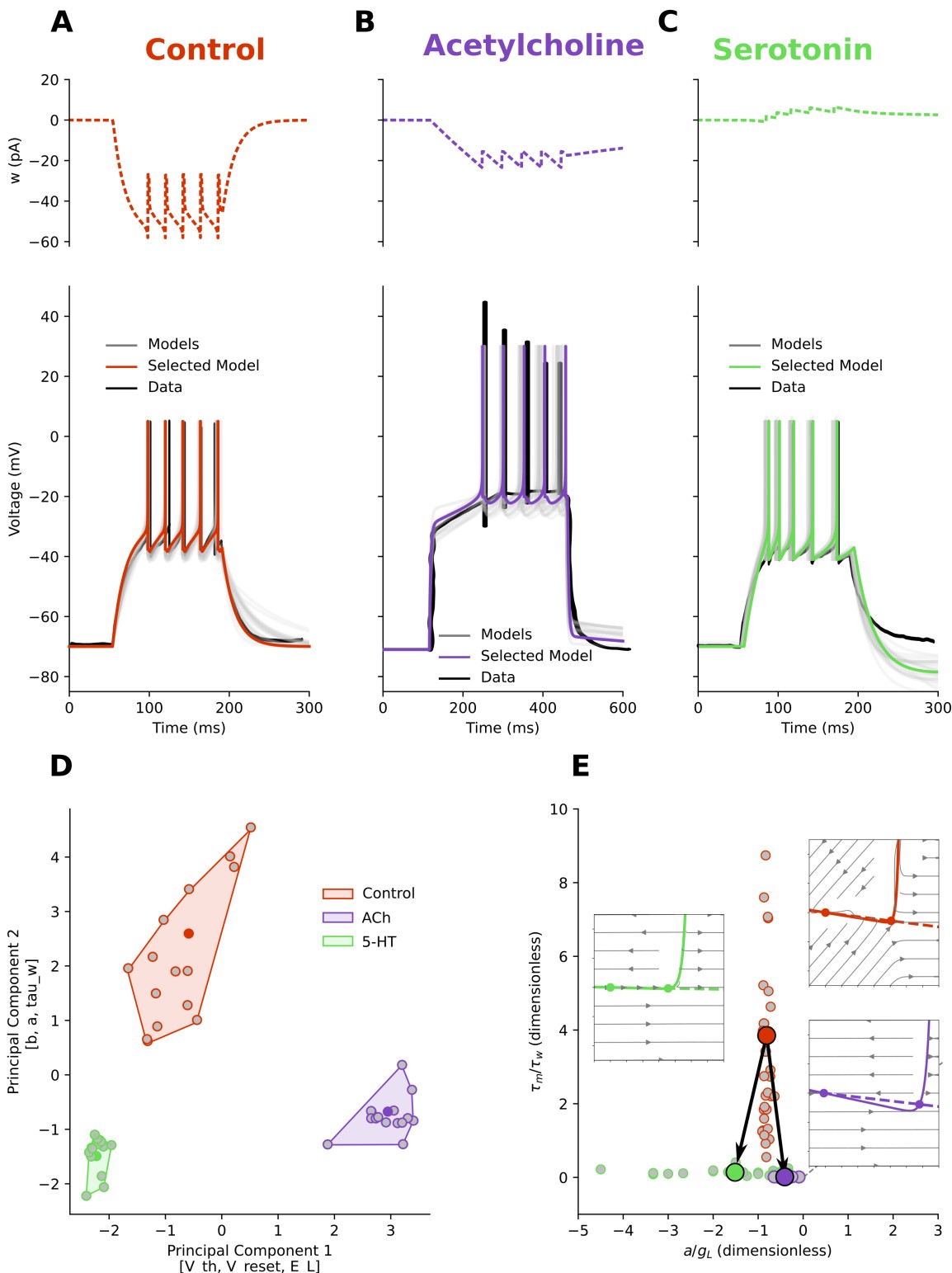

**Fig 7. Neuromodulators speeds cerebellar adaptation and shifts excitability toward the Class II regime. A:** In control conditions, GCs have a delayed time to first spike (TFS, ~ 50 ms) and do not adapt. **B:** In Acetylcholine (ACh) conditions, the TFS doubles (~ 100 ms), and the steady potential rises by ~ 20 mV. **C:** Serotonin (5-HT) conditions are similar to control conditions, with decreased TFS and increased adaptation. **D:** In the PCA-reduced space, with respect to the control (red) cluster, both modulator clusters were shifted to lower values over the first Principal Component [V_th, V_reset, E_L], and the second Principal Component [b, a, tau_w]: ACh (violet) on the right, and 5-HT (light green) on the left. They were all distinct

(Silhouette score $\sim 0.77$). **E**: Excitability landscape. ACh scales the mean downward (smaller $R_\tau$), consistent with slower adaptation relative to the membrane; 5-HT adds a leftward component (smaller $R_a$), indicating weaker subthreshold adaptation relative to the leak. Small circles: individual fits; filled symbols with black arrows: condition means and mean Control→Modulated displacement. Dashed line: Bogdanov–Takens boundary. Insets show phase planes.

components. This combined change in conductance and nullcline geometry scale the neuron toward a Class II (Andronov-Hopf) excitability regime.

5-HT has an opposite effect to ACh on Granule cells (Fig 7C, green point and shaded area in D). It increases the excitability by lowering the left branch of the $V$-nullcline (solid line in inset in Fig 7E), and flattening the $w$-nullcline (dashed line in inset in Fig 7E). Moreover, it also shifts the $V$-nullcline vertex to more hyperpolarised values, producing a spiking adaptation effect.

In parameter space (Fig 7D), control (orange) and both neuromodulators form well-separated clusters. PC1 here reflects the threshold/reset/rest block [$V_{th}, V_{reset}, E_L$], while PC2 is dominated mainly by subthreshold-adaptation [$b, a, \tau_w$]. Projecting the same fits onto the excitability landscape (Fig 7E) reveals how these multi-parameter shifts translate into dynamical regimes. Relative to control, ACh brings the the mean downward (smaller $R_\tau$) with a smaller horizontal change, close to switching. 5-HT produces a stronger leftward shift (smaller $R_a$), indicating a reduction of adaptation strength relative to the leak, and inducing a scaling effect.

### A proof of principle: Estimating the effect of dopamine on human cortical pyramidal neurons

The seminal work by [15] did not explore the effect of Dopamine (DA) on human cortical neurons. Here, we present a proof of principle illustrating how our framework could be used to estimate potential dopaminergic effects in humans. Specifically, we used data from [44] (their Fig 3a1 and 3a2), which report the effect of DA on rodent cortical neurons, mapped those changes onto the AdEx parameters, and applied the same parameter shifts to the average human control model.

We fitted the AdEx model parameters to match control and DA conditions (Fig 8A and B, respectively). Five selected parameters, and how they change under dopaminergic modulation, are illustrated in Fig 8C. The estimated human DA parameters (light blue crosses) were obtained by transferring the rodent DA-induced parameter shifts to the human model (see S1 Appendix). While we acknowledge the possibility of species-specific differences, we found that the firing frequency curves under control conditions in both rodents and humans exhibit similar slopes (Fig 8F, orange curves). This similarity suggests that the baseline excitability properties are comparable, supporting the rationale for this cross-species extrapolation.

Based on this exploratory approach, our results suggest that also in humans DA could increase $g_L$, $V_{th}$, and $V_{reset}$ and decrease $E_L$, $b$, $\tau_w$, and $\Delta_T$. The parameter $a$, which controls the subthreshold adaptation, displays opposite baseline values across species. In rodent control models, $a$ has a negative average value, and it further decreases under dopaminergic conditions. In contrast, the human control model exhibits a positive average value. This difference opens two possibilities when estimating the dopaminergic effect. We can take into account the increase in the absolute value of $a$ or a signed decrease (see Methods for details and S4 Fig). Both approaches produce comparable results in terms of estimated dynamics, while the increase in firing frequency is more pronounced in the second case. In the main text (Fig 8), we report the simulations using the increase in $a$ (absolute value strategy) while in the S4 Fig we also report the case with a decrease in $a$ (signed value strategy).

### Discussion

In this study, we interpret how neuromodulators reshape neuronal excitability by altering the dynamical parameters of a conductance-based model, the Adaptive Exponential Integrate-and-Fire (AdEx). Following the steps of our framework, we

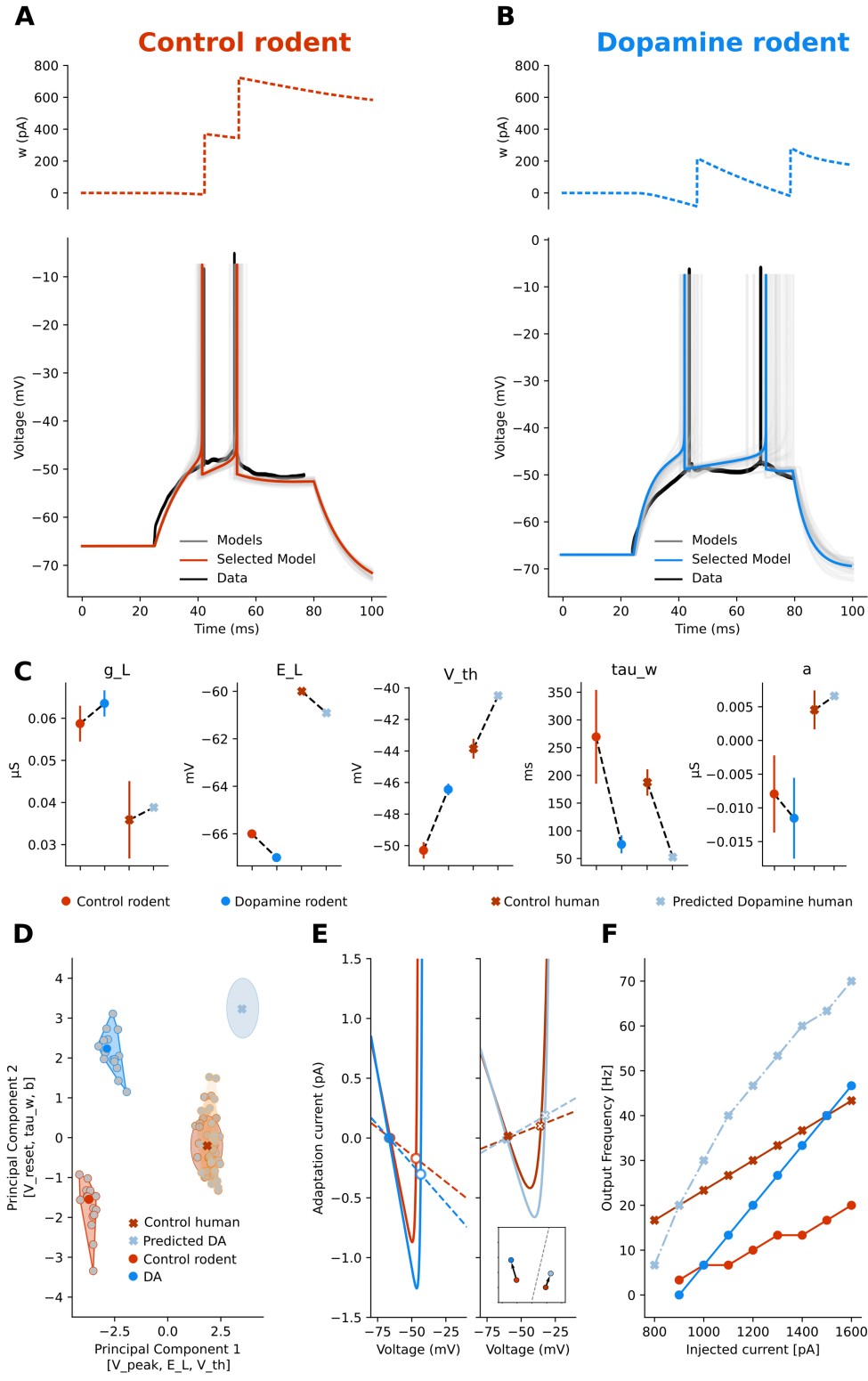

**Fig 8. Estimating dopamine effects on humans.** AdEx model fitting of voltage traces recorded from rodent cortical neurons in control **A.** and after bath application of DA **B**. The top panels show the evolution of the adaptation current $w$, while the bottom panels show the model voltage traces compared to experimental data. Specifically, thin grey lines represent the pool of fitted models, colored lines the selected best model and black traces are the recorded data. **C:** Comparison of selected parameters across conditions and species. In red and blue circles, rodent control and DA average values are plotted, in dark red crosses the average of human control data, and in light blue crosses the estimated human parameters obtained by applying rodent

DA-induced parameter shifts. **D:** PCA of model parameters across conditions. The first Principal Component is mostly influenced by [`V_peak`, `E_L`, `V_th`] while the second Principal Component axes by [`V_reset`, `tau_w`, `b`]. Controls and DA models form distinct clusters, with DA mainly shifting them along the y-axis. **E:** Phase planes in rodent (left) and human (right). DA increases the magnitude of the slope of adaptation. In the inset, the excitability landscape shows that in rodent the DA causes a left–upward shift, reinforcing the neuron's stability within the same excitability class as Control. Similarly, in humans, DA predicts maintenance of the control excitability class. **F:** Firing-frequency curve for rodent and human models in control and DA conditions. The slopes of the control and modulated *f-I* curves in both species are comparable.

first collected electrophysiological data from published papers and extracted features for seven types of neurons from five areas of the human and rodent central nervous system, under the effect of five neuromodulators. Second, we described this data using the AdEx model. Third, we applied dimensionality reduction techniques to study the parameter space of the control and neuromodulated conditions. Fourth, we examined the dynamics of each neuromodulator through bifurcation and phase-plane analyses. Taken together, these steps establish a structured methodology that is both systematic and flexible.

Feature extraction and parameter space analysis revealed that neurons from the same condition, despite differences in their measured features (S6 Fig), were consistently assigned to the same cluster (S3 Fig). This indicates that the framework is robust to inter-trace variability and acts as a filter that reveals the underlying organisation of neuronal excitability. The dimensionality reduction also highlighted the most influential parameters for each neuromodulator, grouping them into distinct, interpretable, and predictive clusters. At the same time, the dimensionality reduction shows the specificity of brain regions. In fact, intrinsic cell properties, such as morphologies, ion channel expression, and connectivity, are fundamentally constrained by region, thus neuromodulators, while reshaping excitability, do not override those properties (S5 Fig).

A conceptual strength of our work lies in our decision not to propose a single "best" model, but rather an ensemble of parameter sets that match the data within acceptable bounds (see S1 Appendix). This is particularly important in cases where only a single electrophysiological trace is available, as is often the case with historical literature. Multiple parameter sets can reproduce the same trace but respond differently under varying inputs. Thus, having a diverse set of plausible models allows for more robust simulation of neurons at the network level and enables progressive refinement as more data becomes available. For instance, if additional traces with different input amplitudes were acquired, the ensemble could be re-ranked or pruned without restarting the fitting process. Importantly, the distribution of parameters across the ensemble also provides insight into their identification. In particular, the average of each parameter across models serves as a reliable indicator of its likely value. We used very broad sets of intervals for the starting AdEx parameters in our fitting. This is the worst possible case to encounter and still produces reasonably good results. In more realistic scenarios, prior information suggesting narrower intervals would further improve accuracy.

The excitability landscape then serves as a lens to interpret these results. Across all areas — cortical, striatal, hippocampal, thalamic, and cerebellar neurons — we found that neuromodulators changed the excitability landscape of neurons in ways that could be compactly captured by our framework. Neuromodulation has two general effects: "switching" the spiking behaviour (between Class I and Class II, partially related to switching between tonic and bursting behaviours), or "tuning" the strength of an existing behaviour. In "switching" instances, the transition between regimes can be described as a shift across the boundary separating saddle-node on invariant circle (SNIC, class I) from Hopf (class II) excitability. This neuromodulatory effect is most prominently induced by Acetylcholine and Histamine (Figs 2, 3, 4, 5, and 7). Such transitions were observed across brain regions and preparations, indicating that the AdEx parameterisation provides a robust lens to compare otherwise heterogeneous datasets. In "tuning" instances, there is no crossing of the boundary but a shift within the same subspace, which changes the occurrence of characterising features. This effect is exerted most clearly by Dopamine and occurs independently of the specific receptor types expressed by the neurons (Figs 3 and 8). Norepinephrine typically exerts switching effects but acts as a tuner in cortical pyramidal neurons (Figs 2, 4, and 5).

Likewise, Acetylcholine, which otherwise causes switching transitions, tunes activity in thalamic reticular neurons without inducing a regime change (Fig 6). The dividing line between Class I and Class II excitability is set by a balance between three parameters (see Materials and methods, Excitability landscape): how strong subthreshold adaptation is ($a$), how quickly that adaptation acts ($\tau_w$), and how much charge the membrane can store ($a\tau_w = C_m$). Many neuromodulators weaken adaptation and/or make it act faster. When that happens, the balance shifts toward Class I: firing can start at arbitrarily low rates and near-threshold spikes take a bit longer to appear. By contrast, simply changing the leak conductance ($g_L$) tends to slide cells along the dividing line without truly changing side. This is why cells that already sit in Class I often move closer to the boundary under modulation but usually don't cross it. To push a cell from Class I into Class II it is needed the opposite move — stronger and/or slower adaptation — which common neuromodulatory pathways are less effective at producing. Hence we frequently observe transitions from Class II to I, but the reverse is comparatively rare.

These findings are particularly interesting because neuromodulation dysfunction is often a key factor in brain disorders. For example, the two most common neurodegenerative diseases, Alzheimer's and Parkinson's, are associated with a decline in the cholinergic system and dopamine deficiency, respectively [49,50]. In depression, imbalances in serotonin and norepinephrine are thought to contribute to symptoms [51,52,66,67], while schizophrenia is linked to dysfunction of the dopaminergic system [53]. Consistent with these associations, many psychiatric treatments act on neuromodulatory systems, such as selective serotonin reuptake inhibitors (SSRIs) for depression, 5-HT$_{2A}$ and D2 receptor antagonists for psychosis, and dopaminergic stimulants for attention-deficit/hyperactivity disorder (ADHD).

In this context, a key insight from the striatal neuron modelling (Fig 3) is relevant: dopamine modulation increases the separation between the parameter sets of direct-pathway (dSPNs) and indirect-pathway (iSPNs) spiny projection neurons. This suggests that dopamine not only modulates individual excitability but also amplifies the functional distinction between these two pathways. Hence, an intriguing future direction would be to test whether such divergence collapses in disease models—such as Parkinson's—where dopaminergic input is impaired. If the dSPN and iSPN parameter distributions were to converge (resulting in a negative silhouette score in PCA), it might indicate a mechanistic biomarker for pathological states.

While the AdEx model is not designed to capture the detailed spike shape, we mitigated this limitation through visual inspection of the top-ranked models. This hybrid approach—quantitative fitting guided by a set of core features, followed by qualitative validation—ensured that selected models faithfully replicated the key dynamics of the source recordings. This methodological flexibility is particularly useful when raw data is unavailable and digitised traces from publications must suffice.

We also emphasise the technical contribution of this work: all modelling and simulations were conducted in Python, using custom scripts for AdEx simulation, feature extraction, parameter space investigation, dimensionality reduction, and visualisation. We deliberately chose not to use established simulation frameworks like NEST [45] or Brian 2 [46], enabling full control and transparency over the parameter exploration and optimisation process, while avoiding additional dependencies. This choice was also didactic, keeping the implementation lightweight and fully accessible.

However, there are also important limitations to our work. The reliance on published traces introduces variability, diverse recording temperatures, and heterogeneous stimulus protocols. Some fits relied on data obtained with agonists rather than endogenous neuromodulator release, which may not fully capture physiological conditions. Furthermore, the AdEx model, while versatile, collapses many ionic processes into abstract parameters. Finally, our framework is limited to single-neuron dynamics and does not yet address the consequences of neuromodulation at the network level.

One of the practical limitations we encountered during our parameter fitting was the lack of precise information regarding the holding current used to maintain neurons at a specific membrane potential in many experimental datasets. This uncertainty introduces variability in estimating the leak reversal potential ($E_L$). The inability to confidently constrain $E_L$ introduces ambiguity in comparing fitted models, particularly when the holding current differs across experimental setups.

Another point is the lack of correction for liquid junction potential in the source data we analysed. This systematic oversight, common across many electrophysiological studies, can result in voltage offsets that propagate through the fitting

and bias parameter estimations of $E_L$ and $V_{th}$. While not critical for comparisons within the same study, it warrants caution when comparing across datasets or interpreting absolute parameter values.

Moreover, in many instances, the neuromodulator used in the experiments was a pharmacological agonist rather than the native molecule. For example, methacholine, a non-specific cholinergic agonist, was used instead of acetylcholine. While such substitutions are standard in experimental neuroscience, they must be kept in mind during interpretation as pharmacological agents may have broader or narrower receptor targets than endogenous ligands.

From a broader perspective, our work provides a bridge between experimental electrophysiology and computational modelling. While detailed Hodgkin-Huxley models offer biophysical fidelity, their complexity makes them less amenable to comparative analyses across neuron types and modulatory states. The AdEx model, by contrast, offers a tractable yet expressive framework that can be scaled to network simulations, subjected to mathematical analysis, and flexibly fitted to empirical data.

Our approach shares conceptual similarities with Bayesian inference. The weighted absolute percent error we use as an objective function can be interpreted as corresponding to a Laplace log-likelihood, where the feature-specific weights implicitly define the covariance structure. Under a uniform prior, this would make the posterior proportional to that likelihood. In our current implementation, however, the retained parameter sets are treated as equally plausible rather than being weighted by their likelihood, so the resulting distribution resembles an unweighted approximation of the posterior. Using a squared error metric and weighting models by their relative accuracy would bring the procedure even closer to a Bayesian estimation scheme.

The implications of this work extend beyond single-neuron modelling. By providing compact descriptions of neuromodulatory effects, we pave the way for neuromodulation-aware network simulations, where different brain states (e.g., sleep vs. wakefulness, attention vs. drowsiness, motor execution vs. rest) can be modelled by adjusting parameter clusters in a biologically plausible way. Furthermore, our framework may support future studies on plasticity and learning, since neuromodulators are known to influence synaptic efficacy and circuit reconfiguration over time. The approach is extensible, generalizable, and particularly well-suited to the kinds of sparse, heterogeneous data that often characterise neuroscience research.

## Conclusion

This study provides a structured and flexible methodology to capture the effects of neuromodulators on neural excitability in the central nervous system. We present a general framework to study how different families of neuromodulators affect neuronal dynamics, and to estimate their effects based on data from other species. Dimensionality reduction analysis reveals that neuromodulation organises models into distinct clusters, allowing us to identify the most influential parameters for each neuromodulator. Finally, the excitability landscape analysis reduces several seemingly different neuromodulatory effects to two broad groups – "switching" and "scaling" – allowing us to show how neuromodulation reshapes the landscape of neuronal dynamics.

## Supporting information

**S1 Fig.** Comparing NEST, Brian 2 and our implementation when using models from [14].
(PDF)

**S2 Fig.** Best 16 cortical pyramidal neuron models in control condition.
(PDF)

**S3 Fig.** Comparison of AdEx model parameters fitted to three electrophysiological traces from the same layer III human cortical neuron.
(PDF)

**S4 Fig.** The effect of dopaminergic modulation on the subthreshold adaptation parameter.
(PDF)

**S5 Fig.** Summary PCA color-coded by brain region.
(PDF)

**S6 Fig.** Summary features-PCA.
(PDF)

**S1 Table.** Extended metadata of electrophysiological recordings across brain regions, species, and neuromodulators.
(PDF)

**S1 Appendix.** Comparing AdEx parameters.
(PDF)

## Acknowledgments

We thank Professor Jeanette Hellgren Kotaleski for her valuable comments on the first version of the manuscript. We thank Belén Montenegro Viñas for being the first user of the pipeline and for providing valuable feedback that helped us improve its usability.

## Author contributions

**Conceptualization:** Ilaria Carannante, Domenico Guarino, Alain Destexhe.

**Data curation:** Ilaria Carannante, Domenico Guarino.

**Formal analysis:** Ilaria Carannante, Domenico Guarino.

**Funding acquisition:** Alain Destexhe.

**Investigation:** Ilaria Carannante, Domenico Guarino.

**Methodology:** Ilaria Carannante, Domenico Guarino.

**Project administration:** Alain Destexhe.

**Software:** Ilaria Carannante, Domenico Guarino.

**Supervision:** Alain Destexhe.

**Validation:** Ilaria Carannante.

**Visualization:** Ilaria Carannante.

**Writing – original draft:** Ilaria Carannante, Domenico Guarino.

**Writing – review & editing:** Ilaria Carannante, Domenico Guarino, Alain Destexhe.

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
