## [Decision Letter · Decision Letter 0]

27 Jul 2025

PCOMPBIOL-D-25-01115

Neuromodulators control neuronal dynamics through feature space reshaping

PLOS Computational Biology

Dear Dr. Carannante,

Thank you for submitting your manuscript to PLOS Computational Biology. After careful consideration, we feel that it has merit but does not fully meet PLOS Computational Biology's publication criteria as it currently stands. Therefore, we invite you to submit a revised version of the manuscript that addresses the points raised during the review process.

Please submit your revised manuscript within 60 days Sep 26 2025 11:59PM. If you will need more time than this to complete your revisions, please reply to this message or contact the journal office at ploscompbiol@plos.org. Please include the following items when submitting your revised manuscript:

We look forward to receiving your revised manuscript.

Kind regards,

Hermann Cuntz

Academic Editor

PLOS Computational Biology

Lyle Graham

Section Editor

PLOS Computational Biology

**Journal Requirements:**

- TM on page: 10.

4) We notice that your supplementary Figures are included in the manuscript file. Please remove them and upload them with the file type 'Supporting Information'. Please ensure that each Supporting Information file has a legend listed in the manuscript after the references list.

6) Please send a completed 'Competing Interests' statement, including any COIs declared by your co-authors. If you have no competing interests to declare, please state "The authors have declared that no competing interests exist". Otherwise please declare all competing interests beginning with the statement "I have read the journal's policy and the authors of this manuscript have the following competing interests"

**Reviewers' comments:**

Reviewer's Responses to Questions

**Comments to the Authors:**

Reviewer #1: See attachment.

Reviewer #2: The manuscript analyses a set of 30 voltage traces collected from the literature on the effect of neuromodulators on different cell types in human or rodent models. The study extracts six features related to spike timing from the voltage traces, and presents a set of Adaptive Exponential Integrate-and-Fire models that fit the extracted features. The authors show that the model parameters populate overlapping parameter spaces for voltage traces from the same neuron under the same condition, but that the application of neuromodulators consistently leads to shifts in the parameter space. The best-fitting model is used to evaluate the phase plane that underlies the dynamics based on the nullclines. Finally, the parameter shift observed for rodents under Dopamine is translated to human neurons under control conditions, to predict the voltage trace for Dopamine.

The topic is interesting and the manuscript has potential, though it could still be improved regarding both analysis and presentation. The manuscript states that it aims to understanding neuromodulators, while at the moment it only brings the complexity of neuromodulation to another level, the parameter space, without sorting neuromodulatory effects. The approach to fit models to study the effect on the dynamics across cell types and species, the use of a set of models instead of a single one, and the detail of the model is appropriate for the research question. However, in my opinion the manuscript does not yet take full advantage of its theoretical analysis. In particular, it does not yet include a detailed comparison of different cell types, neuromodulators or species. Furthermore, the analysis does so far not convincingly show that the fit of the AdEx model or the phase plane analysis provide additional, useful insights as compared to effects of neurotransmitters already identified experimentally (as stated in the manuscript).

About the presentation:

The manuscript has detailed information on all aspects of the study. I appreciate that the methods are very detailed, but the language used is more appropriate for didactic purposes than for a scientific publication. I would recommend to publish this text as explanation for the github, and aim in the manuscript for a more concise presentation – all methods are standard, and can thus be presented substantially shorter. This allows readers to catch the essential steps more quickly.

Also the presentation of the results uses lots of words. A more concise presentation of the results might profit from a table, in which the results on different cells are summarized. At least it would be helpful to sort effects on neuromodulators by effect shared by all or most neurons, and state those once for all, and a more detailed discussion of where differences occur, and how those could be explained.

About the reported results:

The authors have decided to fit selected feature of the voltage traces, instead of the voltage traces itself. Could you please give a more detailed explanation of why that was done? Why fitting only six features, instead of the spike time of every spike? The fits that are shown nevertheless overlap largely with the voltage traces, how did you ensure this overlap if the voltage value was normally not fitted?

The conclusion states the results that allowed to "identify the most influential parameters for each

neuromodulator.", line 819. Do the authors refer to the components most influential for the PCA dimensions? It would be nice to summarize the findings. A sensitivity analysis of the models could provide additional support for such a statement.

PCAs were done independently for each Figure, which makes comparisons impossible. Could the authors also provide one summary PCA plot in which all control traces are compared with all neuromodulatory traces?

Do features already fall into different categories? This would be an important question to answer that helps to understand how much more information the AdEx model fit provides as compared to the features space alone. Are changes in feature space and changes in parameter space correlated?

The AdEx neuron model can show different dynamics at spike onset, compare your Ref. [16], https://doi.org/10.1007/s00422-008-0267-4. Have the authors checked which bifurcation type underlies the onset of spiking? This would be a valuable information.

The paper presents a prediction of neuronal voltage traces for Dopamine. For me this prediction would be only convincing if given a solid motivation. A possible motivation for this prediction could be based on the analysis of the other cells:

* The construction of the shift in parameters used for this prediction, applied to cells for which the same modulator has been used experimentally, does this correlate with what is observed in the analysis of the other voltage traces? For example, there are four voltage traces in which Acetylcholine is modulated, how do the predicted shifts of those four cases compare?

* You might get clearer results when you fit the modulated voltage traces with a regularization to the control parameters. This allows you to identify the change in parameters that is absolutely critical, in the sense of reached despite the regularization, to reach the modulated state.

* Do modulators change the bifurcation diagram qualitatively?

Why do you focus on comparing species instead of, or additional to, brain areas or cell types?

Are your results stable when changing the number of model fits that are considered? Doubling the number of simulations and then using the 32 best models, do we end up with the same clusters?

For detailed comments, please find appended a pdf with comments.

**Have the authors made all data and (if applicable) computational code underlying the findings in their manuscript fully available?**

Reviewer #1: **No: **A github link is included but I got a 404 when I clicked. Maybe the repo is still private?

Reviewer #2: Yes

PLOS authors have the option to publish the peer review history of their article (what does this mean?). If published, this will include your full peer review and any attached files.

Reviewer #1: No

Reviewer #2: No

**Figure resubmission:**
---

## [Decision Letter · Decision Letter 1]

21 Oct 2025

PCOMPBIOL-D-25-01115R1

Neuromodulators control neuronal dynamics through feature space reshaping

PLOS Computational Biology

Dear Dr. Carannante,

Thank you for submitting your manuscript to PLOS Computational Biology. After careful consideration, we feel that it has merit but does not fully meet PLOS Computational Biology's publication criteria as it currently stands. Therefore, we invite you to submit a revised version of the manuscript that addresses the points raised during the review process.

Please submit your revised manuscript within 30 days Dec 21 2025 11:59PM. If you will need more time than this to complete your revisions, please reply to this message or contact the journal office at ploscompbiol@plos.org. Please include the following items when submitting your revised manuscript:

We look forward to receiving your revised manuscript.

Kind regards,

Hermann Cuntz

Academic Editor

PLOS Computational Biology

Lyle Graham

Section Editor

PLOS Computational Biology

**Additional Editor Comments:**

There are a few important comments from Reviewer 2 that need to be addressed. Also, please choose only one title for the final publication.

**Journal Requirements:**

1) Please ensure that the Title in your manuscript file and the Title provided in your online submission form are the same.

3) Please ensure that the funders and grant numbers match between the Financial Disclosure field and the Funding Information tab in your submission form. Note that the funders must be provided in the same order in both places as well.

**Reviewers' comments:**

Reviewer's Responses to Questions

**Comments to the Authors:**

Reviewer #1: See attachment.

Reviewer #2: The authors have for many points implemented our suggestions, or stated convincingly or acceptably why they refrain from doing so. The authors have improved on the presentation, and their classification better summarizes their findings. Nevertheless, I remain with some comments:

Major:

1. Please review the manuscript and change all mentioning of "class A" and "class B" with "class 1" and "class 2" - or provide a definition/reference.

2. The advantage of fitting voltage traces is that we translate voltage differences into parameters that have a biological meaning. Yet, I am still missing the interpretation of the results with regard to a reasonable biological action of the different neuromodulators. For many of them, mechanistic points of action are known, e.g. how they change the resting potential, how do these known mechanisms fit to the biological picture the parameter changes are drawing?

3. Section "Predicting the effect of a neuromodulator": The authors did not provide evidence that what they call prediction has indeed predictive power. I recommend that the authors relabel this subsection and more carefully talk about "a proof of principle that this approach could predict modulatory effects given sufficient data". Generally, the focus of the paper has shifted away from the prediction, which is also reflected in the title, which is good.

4. The authors state that a prediction of neuromodulation is only possible with similar baseline excitabilities, and hence refrain from additional examples. I find this a weak excuse, as I do not agree with the authors that the cortex human and rodent baseline excitabilities are more similar as compared to striatum iSPN/SPN. The f-I curve is more similar in the latter case when normalizing by the highest value, which would in my opinion be a fair comparison.

Minor:

1. Thank you for the PCA plots including all models. It might be interesting to add arrows that link the center of the untreated case with the treated case to see not only the clustering -which is not evident- but also a systematic shift along the PCA components -- similar as done for the excitability landscape plots, but now for all at the same time. For example, Cerebellum and Thalamus (TRN) neurons (and mayby striatum?) react to MCh/ACh with a right downward shift along the principle components, while Thalamus (TRS) neurons show a shift in the opposite direction - any idea how to explain this? Different baseline features? Given the new Figures S5 and S6, if you connect control and treated locations in PCA space with an arrow, do you see more ordered responses in S5 as compared to S6? This is what you suggest, given that models are a more encompassing account of what happens under neuromodulation?

2. My request for a sensitivity analysis was not sufficiently clear to be understandable. The authors addressed the request by a targeted sensitivity analysis for the parameter a and concluded that within the relevant range of a values, the changes to the model are only quantitative. I would find it more interesting to compare how strongly parameter a changes the model as compared to other parameters. A large change in parameter a might result in the same effect as a much smaller change in a second parameter, but PCA only identifies the parameters with the strongest change, not the strongest impact on selected/relevant model readouts. Here you could generate additional insights and take advantage of the model.

3. Switching and scaling is an interesting distinction to make, and the table shown in the rebuttal letter would be a nice addition to the manuscript. The transition between tonic and bursting is a second category that the authors could consider in this table, as even stated as example in the discussion.

4. I have a few more comments in the appended pdf, which the authors could address if they wanted to, but which are not prerequisite for recommending publication.

**Have the authors made all data and (if applicable) computational code underlying the findings in their manuscript fully available?**

Reviewer #1: Yes

Reviewer #2: Yes

PLOS authors have the option to publish the peer review history of their article (what does this mean?). If published, this will include your full peer review and any attached files.

Reviewer #1: No

Reviewer #2: No

**Figure resubmission:**
---

## [Editor Report · Decision Letter 2]

19 Nov 2025

Dear Dr. Carannante,

We are pleased to inform you that your manuscript 'A unified model library maps how neuromodulation reshapes the excitability landscape of neurons across the brain' has been provisionally accepted for publication in PLOS Computational Biology.

Best regards,

Hermann Cuntz

Academic Editor

PLOS Computational Biology

Lyle Graham

Section Editor

PLOS Computational Biology

---

## [Editor Report · Acceptance letter]

PCOMPBIOL-D-25-01115R2

A unified model library maps how neuromodulation reshapes the excitability landscape of neurons across the brain

Dear Dr Carannante,

I am pleased to inform you that your manuscript has been formally accepted for publication in PLOS Computational Biology. Your manuscript is now with our production department and you will be notified of the publication date in due course.

With kind regards,

Zsofia Freund
